# Spin density wave and van Hove singularity in the kagome metal CeTi$_3$Bi$_4$

Pyeongjae Park [1] ✉, Brenden R. Ortiz [1], Milo Sprague[2], Anup Pradhan Sakhya [2], Si Athena Chen[3], Matthias D. Frontzek [3], Wei Tian[3], Romain Sibille [4], Daniel G. Mazzone [4], Chihiro Tabata [5,6], Koji Kaneko [5,6], Lisa M. DeBeer-Schmitt [3], Matthew B. Stone [3], David S. Parker[1], German D. Samolyuk [1], Hu Miao [1], Madhab Neupane[2] & Andrew D. Christianson [1] ✉

Kagome metals with van Hove singularities near the Fermi level can host intriguing quantum phenomena such as chiral loop currents, electronic nematicity, and unconventional superconductivity. However, to our best knowledge, unconventional magnetic states driven by van Hove singularities—like spin-density waves—have not been observed experimentally in kagome metals. Here, we report the magnetic and electronic structure of the layered kagome metal CeTi$_3$Bi$_4$, where Ti kagome electronic structure interacts with a magnetic sublattice of Ce$^{3+}$ $J_{eff} = 1/2$ moments. Neutron diffraction reveals an incommensurate spin-density wave ground state of the Ce$^{3+}$ moments, coexisting with commensurate antiferromagnetic order across most of the temperature-field phase diagram. The commensurate component is preferentially suppressed by thermal fluctuations and magnetic field, yielding a rich phase diagram involving an intermediate single-**Q** spin-density wave phase. First-principles calculations and angle-resolved photoemission spectroscopy identify van Hove singularities near the Fermi level, with the observed magnetic propagation vectors connecting their high density of states, strongly suggesting a van Hove singularity-assisted spin-density wave. These findings establish kagome metals $Ln$Ti$_3$Bi$_4$ as a model platform where the characteristic electronic structure of the kagome lattice plays a pivotal role in magnetic order.

Materials featuring a kagome lattice have attracted considerable attention due to their intriguing electronic structure and the emergence of novel states of matter[1]. Together with Dirac points arising at the K points[2] and flat bands[3-6], one of the most compelling features of the kagome electronic structure is its van Hove singularities (VHSs) at the M points, whose high density of states (DOS) can drive strongly

correlated phenomena (Fig. 1c)[1,7-10]. Notably, each VHS at three distinct M points in a kagome lattice is predominantly governed by a different sublattice, resulting in a marginal on-site interaction vertex (referred to as sublattice interference)[11]. Combined with inter-site Coulomb interactions, this interference greatly enhances unconventional Fermi surface instability near van Hove filling[8]. Indeed, the presence of VHSs

[1]Materials Science and Technology Division, Oak Ridge National Laboratory, Oak Ridge, TN 37831, USA. [2]Department of Physics, University of Central Florida, Orlando, FL 32816, USA. [3]Neutron Scattering Division, Oak Ridge National Laboratory, Oak Ridge, TN 37831, USA. [4]PSI Center for Neutron and Muon Sciences, 5232 Villigen PSI, Switzerland. [5]Materials Sciences Research Center, Japan Atomic Energy Agency, Tokai, Ibaraki 319-1195, Japan. [6]Advanced Science Research Center, Japan Atomic Energy Agency, Tokai, Ibaraki 319-1195, Japan. ✉e-mail: parkp@ornl.gov; christiansad@ornl.gov

near $E_F$ has been shown to produce novel phenomena in kagome metals, such as charge−density wave (CDW) phases and unconventional superconductivity[7,8,10,12–20].

The integration of magnetism into a kagome metal further enriches the potential for exotic phases. Magnetism diversifies the topological characteristics of the Dirac cone, leading to phenomena such as exotic gap openings[2,21], pair-creation of Weyl points[22–24], and the anomalous Hall effect[2,25,26]. The coexistence of strongly correlated electronic behavior and spin degrees of freedom introduces a similar intriguing question: how does the high electronic DOS at $E_F$ (mutually) influence magnetic properties and generate unique magnetic states? One of the most compelling possibilities is that a nesting instability between VHSs could lead to diverging spin-spin correlations, resulting in a spin-density wave (SDW) state−a modulation of the magnetic moments−analogous to how the instability induces CDW through diverging charge correlations[8,9,12]. Notably, while SDWs have been observed in various magnetic systems[27–29], SDWs driven by the VHSs of the kagome electronic structure remain, to our best knowledge, largely unexplored. An elegant way to address this challenge, while also opening additional avenues for investigation, is to add a distinct magnetic sublattice with well-localized moments between non-magnetic kagome layers, where the latter dominate the electronic structure near the Fermi level and control the collective magnetic

behavior through the Ruderman−Kittel−Kasuya−Yosida (RKKY) mechanism. While many novel phenomena have been noted in magnetic kagome materials[3,5,6,19,21,24–26], layered materials that structurally decouple the magnetic sublattice from the kagome motif can facilitate a more systematic exploration of their interplay.

The RTi₃Bi₄ (R = rare earth) family has recently emerged as a promising platform for pursuing the aforementioned ideas[30–33]. These compounds consist of R ions manifesting localized 4$f$ magnetism and non-magnetic Ti kagome layers responsible for their metallicity (Fig. 1a). While the system appears nearly hexagonal due to the Ti kagome layers, its actual symmetry is orthorhombic due to the R sublattice (*Fmmm* space group). Notably, the magnetic R ions form 1D zig−zag chains along the $a$-axis (Fig. 1a), potentially hosting low-dimensional and anisotropic magnetic behavior[30–35]. Despite this symmetry reduction, the electronic structure near the Fermi level retains key features of the kagome lattice formed by Ti atoms. Recent angle-resolved photoemission spectroscopy (ARPES) and density functional theory (DFT) studies on several RTi₃Bi₄ compounds have revealed flat bands, VHSs, and Dirac points[30,32,35–39], suggesting a potential interplay between spin degrees of freedom and the kagome electronic structure. Additionally, these layered compounds can be thinned down through mechanical exfoliation[32], making them a promising platform for studying the two-dimensional limit of magnetism surrounded by a metallic kagome environment.

Among these compounds, CeTi₃Bi₄ stands out as particularly intriguing. The Ce³⁺ ions exhibit quantum magnetism arising from a well-isolated Kramers doublet ground state ($J_{eff} = 1/2$), as evidenced by the saturation of magnetic entropy at Rln(2) near the antiferromagnetic transition at $T_N = 3.2$ K (Fig. 1d, e, also see Supplementary Fig. 1)[30]. While this aspect is often overshadowed by the heavy fermion behavior typically found in Ce-based metals[40], Kondo physics can be relatively marginal when RKKY interactions dominate[40,41], as is the case in CeTi₃Bi₄ (see Supplementary Note 4 for further explanation). Additionally, the spin−orbital coupled nature of its ground state, combined with the surrounding crystal-field environment, promotes strong magnetic anisotropy[42]. This is evident in the anisotropic *M*−*H* behavior of CeTi₃Bi₄ below $T_N$, with a spin-flop transition observed for **H**//**b** (see Fig. 1f)[30]. Thus, CeTi₃Bi₄ provides a unique platform for investigating the interplay between anisotropic quantum magnetism and kagome electronic structure.

In this study, we present an extensive investigation of CeTi₃Bi₄ utilizing neutron scattering, ARPES, and DFT, and argue that CeTi₃Bi₄ represents a compelling case of unusual magnetic ground states driven by the VHS of the kagome electronic structure. Neutron diffraction measurements reveal an incommensurate SDW ground state of $J_{eff} = 1/2$ magnetic moments coexisting with a commensurate antiferromagnetic spin modulation. Their modulation wave vectors are nearly identical to that associated with the $2a \times 2a$ CDW found in other kagome metals, where the VHS near the Fermi level has been suggested to be important[17–19]. This coexisting phase is observed across a majority of the temperature-field phase diagram, but transforms into a paramagnetic (or polarized) phase via an intermediate single-**Q** SDW state lacking the commensurate spin component, as temperature (or magnetic field) increases. Notably, both the commensurate and incommensurate modulations propagate perpendicular to the Ce³⁺ chains, indicating competing long-range magnetic correlations between well-separated nearest-neighbor (NN) and further-NN Ce³⁺ chains ($d > 5.9$ Å), likely mediated by conduction electrons. Interestingly, ARPES and DFT results identify VHSs near $E_f$ at the M points of the pseudo-hexagonal reciprocal lattice, manifesting high DOS. The observed magnetic ordering wave vectors closely align with their separation vector, suggesting that a nesting instability between these singularities may have driven the observed magnetic ground state.

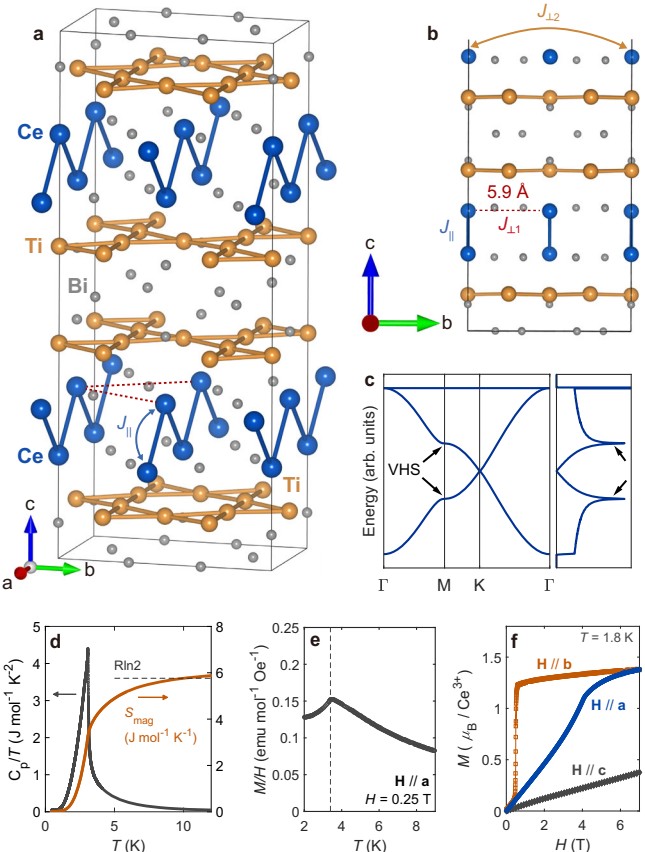

**Fig. 1 | Structure and bulk properties of CeTi₃Bi₄. a** Crystal structure of CeTi₃Bi₄ consisting of Ti kagome layers and 1D chains of Ce³⁺ ions aligned along the $a$-axis. **b** Side views of the crystal structure. Black solid lines in **a**, **b** illustrate the crystallographic unit cell. **c** Schematic electronic structure, dispersion, and density of states (DOS) for a kagome lattice, obtained from a simple 2D tight-binding model. **d** Temperature-dependent magnetic heat capacity of CeTi₃Bi₄ after subtracting non-magnetic components (see "Methods") and corresponding magnetic entropy. **e** Temperature dependence of magnetization (*M*) along the $a$-axis, highlighting an antiferromagnetic transition around 3.4 K. **f** Highly anisotropic field-dependent magnetization in the magnetically ordered CeTi₃Bi₄ ($T < T_N$).

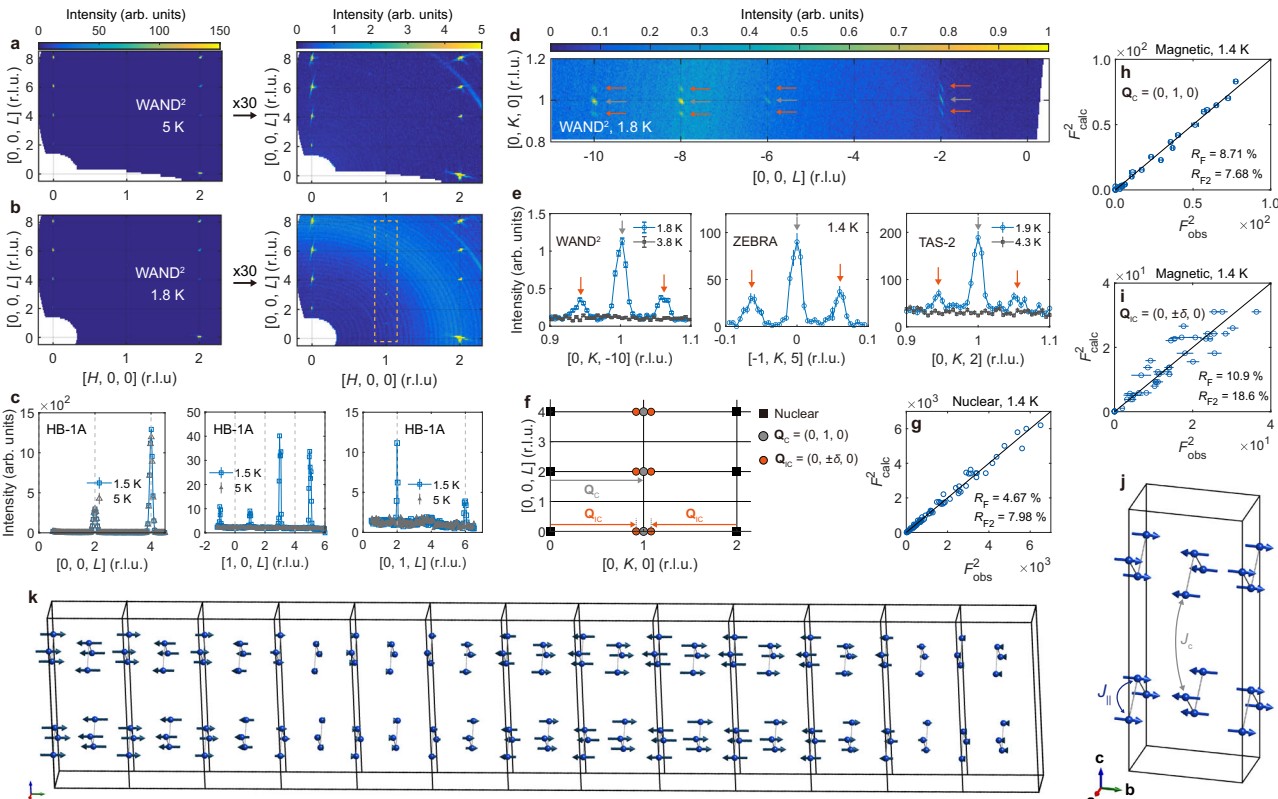

**Fig. 2 | Co-existing commensurate antiferromagnetic and incommensurate spin density wave orders revealed by neutron diffraction. a, b** Single-crystal neutron diffraction profiles of the (*H0L*) plane at 1.8 K ($T < T_N$) and 5 K ($T > T_N$). Right panels of **a**, **b** show the same plot as the left plots but scaled to magnify weak reflections. The enhanced background signal in **b** originates from liquid helium in the cryomagnet's sample space and should not be interpreted as part of the sample's signal. The dashed box in b highlights magnetic reflections. **c** Comparing the diffraction intensity scans along the *L* direction measured at 1.5 K ($T < T_N$) and 5 K ($T > T_N$). **d** Diffraction profiles on the (*0KL*) plane at 1.8 K. **e** Intensity scans along the *K* direction around a commensurate magnetic reflection, each collected from different beamlines and crystals. **f** Two magnetic ordering wave vectors that describe

the observed magnetic Bragg peaks: commensurate $\mathbf{Q}_C = (0, 1, 0)$ and incommensurate $\mathbf{Q}_{IC} = (0, \delta, 0)$ in reciprocal lattice units. Orange (gray) arrows in **d**–**f** point out magnetic reflections associated with $\mathbf{Q}_{IC}$ ($\mathbf{Q}_C$). Note that $\delta$ is temperature-dependent (see Fig. 3). **g** Least-square refinement result of 74 nuclear reflections collected at ZEBRA. Error bars of the observed intensities are much smaller than data symbols. **h, i** Least-square refinement results of 24 commensurate and 47 incommensurate magnetic reflections collected at ZEBRA, respectively. **j, k** Magnetic structure solutions obtained from the analysis result in (**h**, **i**), respectively. In both solutions, the magnetic moments are uniaxially aligned to the *b*-axis. Error bars in **c** and **e** (**h, i**) represent the uncertainty in the measured intensity (fitted integrated intensity).

## Results and discussion

### Single-crystal neutron diffraction

The antiferromagnetic order in CeTi$_3$Bi$_4$ was determined via single-crystal neutron diffraction measurements conducted at multiple diffractometers (see "Methods"). All diffraction data in this study are referenced to the reciprocal lattice of the conventional orthorhombic unit cell (Fig. 1a; also see Supplementary Note 1). Figure 2a–c shows the diffraction profile of CeTi$_3$Bi$_4$ across a broad region of reciprocal space, both below and above $T_N$. Above $T_N$ (Fig. 2a, c), nuclear reflections are observed at $\mathbf{Q}_{nuc} = (h, k, l)$ with all-even or all-odd indices, following the selection rule of the FCC structure. Below $T_N$, additional Bragg reflections appear at $\mathbf{Q} = \mathbf{Q}_{nuc} + (0, 1, 0)$, approximately two orders of magnitude weaker than the nuclear reflections (Fig. 2b, c). These signals become weaker at higher momentum transfer (Supplementary Fig. 5), which, along with their presence only below $T_N$, confirms their magnetic origin.

Interestingly, diffraction profiles along the $(0, k, 0)$ direction reveals additional incommensurate reflections near $\mathbf{Q} = \mathbf{Q}_{nuc} + \mathbf{Q}_C$ below $T_N$, where $\mathbf{Q}_C = (0, 1, 0)$ is the commensurate magnetic modulation described above. At 1.8 K, these reflections are separated from $\mathbf{Q} = \mathbf{Q}_{nuc} + \mathbf{Q}_C$ by $(0, \pm0.06, 0)$ (r.l.u.), indicating a long-period spin modulation with a wavelength of $\sim17|\mathbf{b}| \simeq 17.5$ nm. Notably, these incommensurate reflections were consistently observed across

multiple measurements using three different crystals and diffractometers (Fig. 2e). Like the commensurate magnetic reflections associated with $\mathbf{Q}_C$, these incommensurate reflections weaken at higher $\mathbf{Q}$, as expected for magnetic scattering (Supplementary Fig. 5). Together these observations suggest that CeTi$_3$Bi$_4$ hosts two intrinsic magnetic modulations in the ordered phase—one commensurate ($\mathbf{Q}_C$) and one incommensurate [$\mathbf{Q}_{IC} = (0, \pm0.94, 0)$]—which is not likely due to sample inhomogeneity or crystal mosaic as they would result in measurement-dependent results (also see Supplementary Fig. 2). Figure 2f illustrates the schematic diffraction pattern from these two ordering wave vectors, while Supplementary Note 2 provides additional explanation about assigning $\mathbf{Q}_{IC} = (0, \pm0.94, 0)$ instead of $(0, \pm0.06, 0)$.

Further analysis of the magnetic Bragg peak intensities clearly reveals the two spin configurations associated with $\mathbf{Q}_{IC}$ and $\mathbf{Q}_C$, respectively. A key observation is the absence of both $(0, 1, 0)$ and $(0, 1 \pm0.06, 0)$ magnetic peaks, confirmed by multiple neutron diffraction measurements (e.g., see Fig. 2d). This indicates that both spin modulations consist solely of magnetic moments aligned along the *b*-axis, consistent with the easy-axis anisotropy along this direction, as evidenced by the *M*–*H* curves in Fig. 1f. Importantly, the only way to incorporate the incommensurate modulation of $\mathbf{Q}_{IC}$ into this uniaxial spin configuration is by introducing a modulation of the local moment

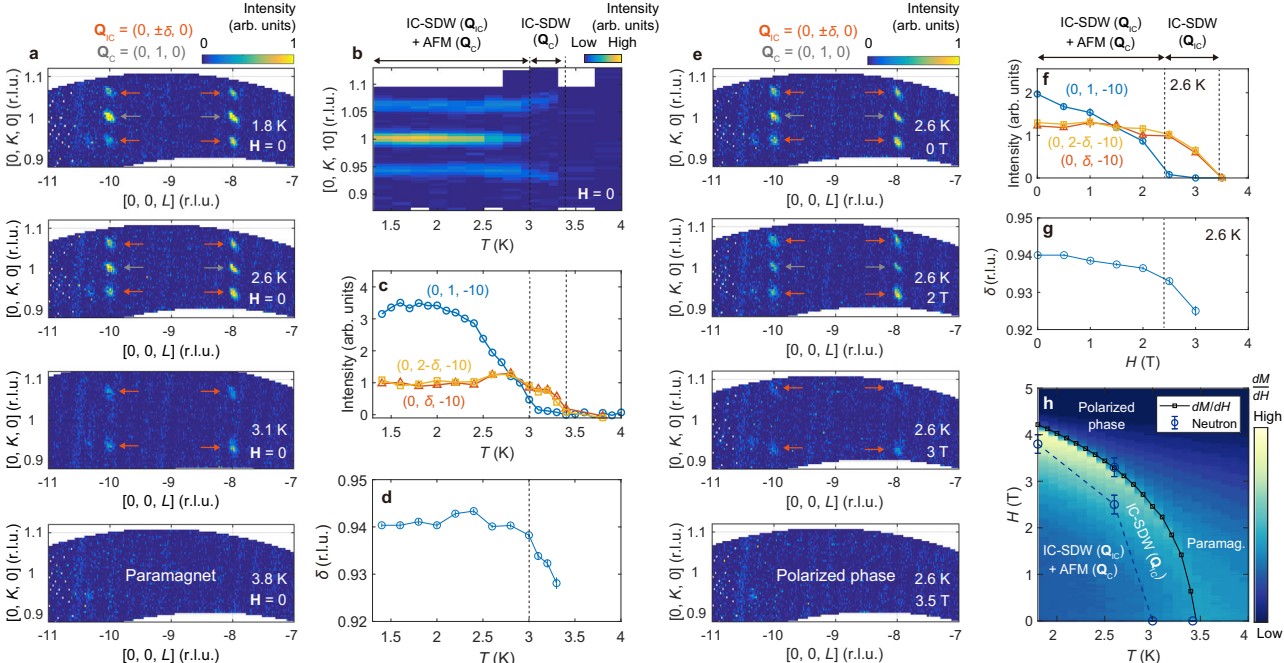

**Fig. 3 | Temperature and field dependence of the two distinct spin modulations. a**, **b** Temperature-driven evolution of commensurate ($\mathbf{Q}_C$, gray arrows) and incommensurate ($\mathbf{Q}_{IC}$, orange arrows) magnetic reflections. **c** Temperature-dependent integrated intensities of the $\mathbf{Q}_c$ and $\mathbf{Q}_{IC}$ peaks at/around (0, 1, −10). **d** Temperature dependence of $\delta$ for the incommensurate modulation $\mathbf{Q}_{IC}$ = (0, $\delta$, 0). The two-step phase transition process at $T_N \approx 3.4$ K and $T_2 = 3.0$ K is evident throughout (**a**–**d**). **e** Field-driven evolution of commensurate ($\mathbf{Q}_C$, gray arrows) and incommensurate ($\mathbf{Q}_{IC}$, orange arrows) magnetic reflections at $T = 2.6$ K. **f** Field-dependent integrated intensities of the $\mathbf{Q}_c$ and $\mathbf{Q}_{IC}$ peaks at/around (0, 1, −10).

**g** Field dependence of the ordering wave vector component $\delta$. **h** Overall temperature-field ($\mathbf{H}$ // **a**) phase diagram summarizing the neutron diffraction measurement results. A color code shows $dM/dH$ extracted from multiple $M$–$H$ curves measured in the range of 1.8 K < $T$ < 4 K with a 0.1 K step. The fine-step phase boundary between the polarized and intermediate single-$\mathbf{Q}$ phases was determined using these $dM/dH$ curves. All neutron diffraction data shown in this figure were collected at WAND[2]. Error bars in **c** and **f** (**d**, **g**) represent the uncertainties in the fitted integrated intensity (fitted peak position).

length, i.e., a SDW-type order. Additionally, the zero components of $\mathbf{Q}_{IC}$ and $\mathbf{Q}_C$ along directions perpendicular to **b\*** suggest that the $Ce^{3+}$ sublattices are ferromagnetically aligned along both the $Ce^{3+}$ chain (// **a**) and out-of-plane (// **c**) directions.

The spin configurations shown in Fig. 2j, k are the only solutions compatible with these findings. This conclusion is further supported by our least-square refinement with 74 nuclear reflections and 24 (47) commensurate (incommensurate) magnetic reflections collected at ZEBRA, PSI (Fig. 2g–i). Indeed, the best agreement was found with the spin configurations depicted in Fig. 2j, k, while the other candidates yielded worse agreement factors ($R_F$ and $R_{F2}$), particularly due to their predicted intensity at $\mathbf{Q}$ = (0, 1, 0) and (0, 1 ± 0.06, 0) (Supplementary Figs. 3 and 4). A similar refinement using data from WAND[2] (HFIR) reached the same conclusion (Supplementary Fig. 6). This conclusively demonstrates a SDW ground state of the $Ce^{3+}$ $J_{eff}$ = 1/2 moments in $CeTi_3Bi_4$, which co-exists with a commensurate antiferromagnetic spin configuration.

The temperature dependence of the two magnetic modulations reveals a close correlation between them and the resultant intriguing phase diagram. Figure 3a shows neutron diffraction profiles around (0, 1, −8) and (0, 1, −10) at various temperatures across $T_N$. While both $\mathbf{Q}_{IC}$ and $\mathbf{Q}_C$ persist up to 2.6 K, only $\mathbf{Q}_{IC}$ is found at 3.1 K. A detailed examination with fine temperature steps reveals two-step phase transitions (Fig. 3b–d; full datasets in Supplementary Fig. 7). The incommensurate modulation $\mathbf{Q}_{IC}$ = (0, ± $\delta$, 0) first emerges below $T = 3.4$ K ≃ $T_N$, with $\delta$ being temperature-dependent (Fig. 3d). This is followed by the appearance of the commensurate modulation $\mathbf{Q}_C$ at $T_2 = 3.0$ K, below which $\mathbf{Q}_{IC}$ and $\mathbf{Q}_C$ coexist. This two-step transition process has been consistently observed across multiple diffraction measurements, confirming its intrinsic nature.

The temperature dependence of $\delta$, or $\delta(T)$, merits further explanation. While $\delta(T)$ exhibits subtle but clear changes when only $\mathbf{Q}_{IC}$ is present ($T_2 < T < T_N$), it locks into a constant value $\delta \simeq 0.94$ below $T_2$ (Fig. 3b, d). This suggests that $\mathbf{Q}_{IC}$ and $\mathbf{Q}_C$ are not independent, implying possible double-$\mathbf{Q}$ nature of the magnetic ground state rather than phase separation. This is further indicated by heat capacity measurements with fine temperature steps, which show two distinct transitions approximately 0.3–0.4 K apart, consistent with neutron diffraction results (see Supplementary Fig. 1). Further comparative analysis of $\mathbf{Q}_{IC}$ and $\mathbf{Q}_C$ based on their temperature-dependent correlation lengths (see Supplementary Note 6 and Supplementary Fig. 8) also suggests an interdependence between these two modulations. Nevertheless, definitive evidence for the double-$\mathbf{Q}$ nature below $T_2$ is still lacking, and we leave it as a future challenge.

Interestingly, the response of $\mathbf{Q}_{IC}$ and $\mathbf{Q}_C$ to an external magnetic field ($\mathbf{H}$) is very similar to their temperature dependence. Figure 3e shows neutron diffraction profiles around (0, 1, −8) and (0, 1, −10) under $\mathbf{H}$ // **a** at various field strengths and $T = 2.6$ K, showing strong similarities to their temperature dependence in Fig. 3a (also see Supplementary Fig. 9). As the field increases, $\mathbf{Q}_C$ disappears first while $\mathbf{Q}_{IC}$ remains nearly intact (0 T < $H$ < 2.5 T in Fig. 3f). Once $\mathbf{Q}_C$ is fully suppressed and the systems only holds a single modulation (single-$\mathbf{Q}$), $\mathbf{Q}_{IC}$ is rapidly suppressed as the field increases further (2.5 T < $H$ < 3.5 T in Fig. 3f). Notably, this suppression of the incommensurate component is accompanied by a faster decrease in $\delta$ (Fig. 2g), similar to its temperature dependence (Fig. 2d).

Figure 3h presents the overall temperature-field ($\mathbf{H}$ // **a**) phase diagram derived from our neutron diffraction and magnetization measurements. While both modulations coexist over a broad range, thermal fluctuations and external magnetic fields suppress $\mathbf{Q}_C$ more

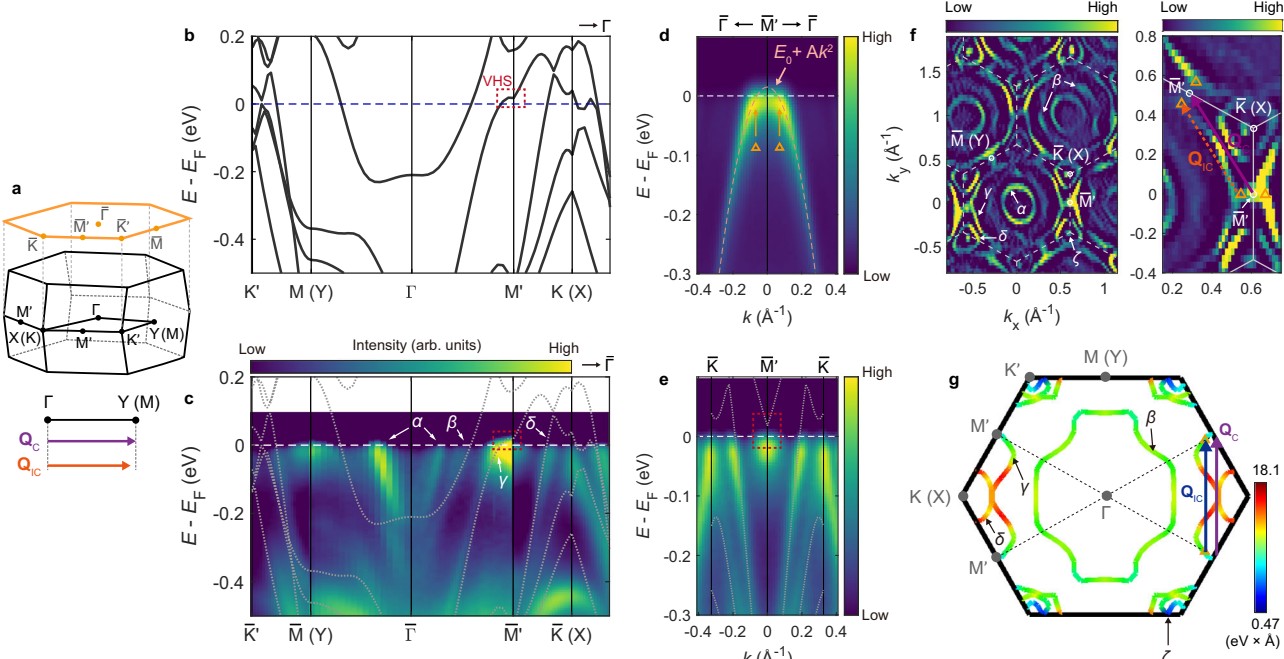

**Fig. 4 | Calculated and measured electronic structure of CeTi₃Bi₄ near the Fermi energy $E_F$. a** First Brillouin zone of CeTi₃Bi₄ in the primitive orthorhombic convention, with labels of key high-symmetry **Q** points. For clarity, the **Q**c and **Q**IC wave vectors are compared with the $\overrightarrow{\Gamma Y}$ wave vector. **b, c** Electronic structure dispersion of CeTi₃Bi₄ obtained from DFT calculations and ARPES measurements. **d** Enlarged ARPES intensity spectrum along the $\bar{\Gamma} - \bar{M}'$ direction with a quadratic dispersion fit (pink dashed line). Orange arrows indicate the positions of maximum intensity. **e** Similar to **d**, but along the $\bar{K} - \bar{M}'$ direction. The spectra in **d, e** are symmetrized with respect to the reflection plane at the $\bar{M}'$ point. DFT calculations (gray dotted lines) are overlaid on the ARPES spectrum in (**c, e**). **f** Fermi surface measured by

ARPES with an enlarged view (right panel) around two adjacent $\bar{M}'$ points, illustrating the (quasi-)connection between VHSs by **Q**c and **Q**IC. The color scale in **f** represents the second derivative of the measured intensity (arb. units), while the original intensity spectrum is shown in Supplementary Fig. 11. An energy integration range of 5 meV was used to construct the ARPES Fermi surface plots. **g** Fermi surface cross-section from the DFT band structure. The color scale indicates the Fermi velocity (eV × Å). Orange and purple arrows in (**f**), or blue and purple arrows in (**g**), indicate the **Q**c and **Q**IC wave vectors, with their length scales synchronized to the background plot in Å⁻¹. All ARPES measurements were conducted using an incident photon energy of 65 eV.

rapidly, resulting in an intermediate incommensurate single-**Q** phase before transitioning to either a paramagnetic or fully-polarized phase. Also, the relative intensity profile of the incommensurate magnetic Bragg peaks (**Q**IC) remains nearly the same throughout the ordered antiferromagnetic phases below $T_N$, with only a uniform intensity scaling due to changes in the size of the ordered moments (see Supplementary Fig. 10). Thus, the SDW-type modulation shown in Fig. 2k persists across the entire antiferromagnetic region in Fig. 3h, highlighting its dominance in CeTi₃Bi₄.

In Ce-based metallic systems, magnetic interactions between Ce³⁺ localized moments are generally mediated via conduction electrons through the RKKY mechanism. Indeed, the potential dominance of the RKKY mechanism in driving the observed magnetic phase diagram (Fig. 3h) is implied by previous studies[43]. In rare-earth metallic magnets with uniaxial anisotropy, two-step paramagnetic−incommensurate−commensurate transitions with temperature have been widely observed (see Table 1 in Ref. 43). This phenomenon has been modeled through competing exchange interactions arising from the long-ranged RKKY interactions at the level of mean-field theory[43]. In this model, the Fourier transformed interaction matrix $J(\mathbf{Q})$ has a minimum at a commensurate wave vector for $T = 0$, meaning that incommensurate spin length modulation is unstable at $T = 0$. However, thermal fluctuations ($T > 0$) shift this minimum to an incommensurate wave vector, leading to intermediate incommensurate ordering between the commensurate and paramagnetic phases. Moreover, an external magnetic field appears to mimic the effect of thermal fluctuations, inducing two-step commensurate−incommensurate−ferromagnetic transitions[43]. This behavior resembles the phase diagram of CeTi₃Bi₄ (Fig. 3h),

suggesting that applying the RKKY framework is appropriate for this system. Supplementary Note 5 provides additional arguments supporting the predominance of the RKKY mechanism in driving the **Q**IC and **Q**c modulations.

Despite the similar phase diagram, a unique feature found in CeTi₃Bi₄ is that the incommensurate SDW phase persists even after the appearance of the commensurate component (**Q**c), unlike the previous cases where incommensurate SDW modulation is intrinsically unstable for $T = 0$[43]. The amplitude of **Q**IC remains significant at the lowest measurement temperature (~1.8 K), as evident in the ordered moment sizes of **Q**IC and **Q**c obtained from least-square refinement (see Supplementary Table 4). This suggests that the conduction electron profile of CeTi₃Bi₄ possesses additional driving forces that favor the incommensurate SDW modulation, which were absent in previous examples[43] of rare-earth metallic magnets. Notably, **Q**C and **Q**IC are nearly identical to the modulation wave vector associated with the $2a \times 2a$ CDW found in other kagome materials[17–19] (=$\overline{\Gamma M}$, see Fig. 4a or Supplementary Note 1), potentially indicating they share a common origin related to the kagome electronic structure.

## Electronic structure analysis

The arguments above led us to explore the Fermi surface topology of CeTi₃Bi₄, as the $J(\mathbf{Q})$ profile shaped by RKKY interactions, which determines the magnetic ground state, should be encoded in the Fermiology. Thus, we analyzed the electronic structure close to $E_F$ using ARPES and DFT (Fig. 4). For the DFT calculations, we utilized the result from LaTi₃Bi₄ as it allows for investigating the uncorrelated electronic structure of $Ln$Ti₃Bi₄ near the Fermi level with significantly decreased complexity of the calculation due to the non-magnetic

nature of La (see "Methods"). Additionally, we adopted pseudo-labels of high-symmetry $\mathbf{Q}$ points from a hexagonal framework (see Fig. 4a), as done in prior RTi$_3$Bi$_4$ studies[30,32,37,38]. See "Methods" for more explanations.

Both the DFT and ARPES measurements reveal that CeTi$_3$Bi$_4$ exhibits VHSs near the Fermi level, $E = E_F$, at the M' points–four of the six M points in a hexagonal framework that do not correspond to the orthorhombic Y point (=$\mathbf{b}$*). Figure 4b, c shows the calculated and measured electronic structures near $E_F$. Except for the α band forming a small pocket around the $\bar{\Gamma}$ point, which likely has surface character (see Supplementary Note 7 and Fig. 13), our DFT band structure with a slight adjustment of $E_F$ to +0.043 eV provides a good description of the ARPES spectrum around $E_F$. Some apparent discrepancies between the ARPES and DFT results below the Fermi energy (Fig. 4c) may stem from using LaTi$_3$Bi$_4$ in the calculations: the validity of this approximation becomes less reliable at lower energies, where the influence of Ce$^{3+}$ electronic states gets more pronounced. Interestingly, strong spectral weight is observed at the $\bar{M}'$ point at the $E_F$ (Fig. 4c), where DFT reveals the presence of a VHS (red dashed box in Fig. 4b). A more careful analysis through the $\bar{\Gamma} - \bar{M}' - \bar{\Gamma}$ (Fig. 4d) and $\bar{K} - \bar{M}' - \bar{K}$ (Fig. 4e) cuts further confirms the saddle point nature of the band dispersion at $\bar{M}'$ near $E_F$ (see also Supplementary Fig. 12). A simple quadratic fit indicates the VHS is positioned at $E_F$ + 0.015 eV (Fig. 4d), very close to $E_F$. Notably, the VHS manifests as a slightly extended line signal along the $\bar{\Gamma} - \bar{M}'$ direction in the Fermi surface geometry (Figs. 4d, f), with the maximum intensity at the $\mathbf{Q}$ positions marked by orange triangles in Fig. 4d, slightly offset from $\bar{M}'$. This could be simply due to the slight upward shift of the VHS from $E_F$ (Fig. 4d), though a similar observation in NdTi$_3$Bi$_4$ was interpreted as higher-order VHS with quartic dispersion[38]. Regardless, this suggests not only a high DOS at $\bar{M}'$ but also around $\bar{M}'$ due to the proximity of the VHS.

Interestingly, the ordering wave vectors $\mathbf{Q}_{IC}$ and $\mathbf{Q}_C$ appear to correspond to the wave vectors connecting the high DOS at adjacent $\bar{M}'$ points (Fig. 4f, g), suggesting relevance between the magnetic order and a nesting instability between VHS in CeTi$_3$Bi$_4$. Due to the large DOS, VHS near $E_F$ can induce CDW or SDW instabilities, with their modulation period determined by a wave vector connecting these points[7–9,12]. In a perfect hexagonal structure, the VHSs at the six M points would result in three equivalent nesting vectors related by three-fold rotational symmetry ($C_{3z}$), leading to the $2a \times 2a$ type tri-directional density-wave modulation. However, in orthorhombic CeTi$_3$Bi$_4$, the VHS only appears at four $\bar{M}'$ points according to DFT (Fig. 4b), leaving a nesting vector aligned with the $\Gamma$–Y direction (// $\mathbf{b}$*) as the only active one. Indeed, $\mathbf{Q}_C = (0, 1, 0)$ ($|\mathbf{Q}_C|$ ~ 0.61 Å$^{-1}$) is consistent with this unique nesting vector, whereas $\mathbf{Q}_{IC} = (0, 0.94, 0)$ ($|\mathbf{Q}_{IC}|$ ~ 0.57 Å$^{-1}$) is slightly shorter (Fig. 4a).

The extended high DOS along the $\bar{\Gamma} - \bar{M}'$ direction, as suggested in Fig. 4d, may explain why $\mathbf{Q}_{IC}$ is as prominent as $\mathbf{Q}_C$: a nesting path connecting the two orange triangles near $\bar{M}'$ could also be significant [orange (blue) arrows in Fig. 4f, g], which is marginally shorter than the ideal vector connecting the $\bar{M}'$ points (=$\mathbf{Q}_C$). Notably, $\mathbf{Q}_{IC}$ nicely agrees with the separation vector of the two orange triangles. More generally, given the extended high DOS along the $\bar{\Gamma} - \bar{M}'$ line (Fig. 4d), any wave vectors between (0, δ, 0) and (0, 1, 0), with δ greater than a certain threshold $\delta_0$ but less than 1, may connect $\mathbf{Q}$ positions near $\bar{M}'$ with high DOS (Fig. 4g). Such a subtle feature, determined by the fine band structure in the vicinity of $\bar{M}'$, may be responsible for the coexistence of $\mathbf{Q}_C$ and $\mathbf{Q}_{IC}$ (Fig. 3h) and the continuous variation of $\mathbf{Q}_{IC}$ by temperatures (Fig. 3d).

Overall, our comprehensive investigation using neutron diffraction, ARPES, and DFT conclusively confirms the incommensurate SDW modulation in the Ce$^{3+}$ $J_{eff} = 1/2$ moments and suggests its relationship to the VHS of Ti-derived Kagome bands. The latter connection warrants more careful investigation, such as a temperature-dependent electronic structure study across $T_N$ and $T_2$. Another important aspect

is the impact of the Ce$^{3+}$ characteristics–strong easy-axis magnetic anisotropy and quantum spin–on the observed ground state. The uniaxial magnetic anisotropy can promote the SDW by disfavoring alternative spin spiral orders without length modulation[43] (see Supplementary Fig. 4), which could also arise from a nesting instability, as observed in NdAlSi[44]. However, if and how the quantum aspect of the $J_{eff} = 1/2$ ground state doublet impacts SDW formation remains unclear based on our current results. It may aid in the SDW modulation in that smaller spin size reduces the energy cost of modulating the ordered moments (which arises from Heisenberg interaction terms) but also exhibits stronger longitudinal quantum fluctuations. Further spectroscopic studies, which are sensitive to longitudinal spin fluctuations, would be illuminating.

## Methods

### Sample preparation
CeTi$_3$Bi$_4$ single crystals are grown through a bismuth self-flux. Elemental reagents of Ce (AMES), Ti (Alfa 99.9% powder), and Bi (Alfa 99.999% low-oxide shot) were combined at a 2:3:20 ratio into 2 mL Canfield crucibles fitted with a catch crucible and a porous frit[45]. The crucibles were sealed under approximately 0.7 atm of argon gas in fused silica ampoules. Each composition was heated to 1050 °C at a rate of 200 °C/hr. Samples were allowed to thermalize and homogenize at 1050 °C for 12–18 h before cooling to 500 °C at a rate of 2 °C/h. Excess bismuth was removed through centrifugation at 500 °C. Crystals are a lustrous silver with a hexagonal habit. The samples are mechanically soft and are easily scratched with a knife or wooden splint. They are layered in nature and readily exfoliate using adhesive tape. Samples with side lengths up to 1 cm are common. We note that samples are moderately stable in air and tolerate common solvents and adhesives (e.g., GE Varnish, isopropyl alcohol, toluene) well. However, the samples are not indefinitely stable and will degrade, tarnish, and spall if left in humid air for several days.

### Bulk property measurements
Magnetization measurements of CeTi$_3$Bi$_4$ single crystals were performed in a 7 T Quantum Design Magnetic Property Measurement System (MPMS3) SQUID magnetometer in vibrating-sample magnetometry (VSM) mode. The crystallographic orientations were carefully determined prior to the measurements using Laue XRD[30]. Samples were mounted to quartz paddles using a small quantity of GE varnish or n-grease. All magnetization measurements were performed under zero-field-cooled conditions unless specified. Heat capacity measurements on CeTi$_3$Bi$_4$ single crystals between 300 K and 1.8 K were performed in a Quantum Design 9 T Dynacool Physical Property Measurement System (PPMS) equipped with the heat capacity option. Additional heat capacity measurements from 400 mK to 9 K utilized a $^3$He insert for the Quantum Design 14 T PPMS. LaTi$_3$Bi$_4$ was used as the reference nonmagnetic species.

### Single-crystal neutron diffraction
Single-crystal neutron diffraction measurements were conducted using the ZEBRA thermal neutron diffractometer at the Swiss spallation neutron source (SINQ) and the triple-axis spectrometer HB-1A at the High Flux Isotope Reactor (HFIR). At ZEBRA, we mount the sample in a Joule-Thomson close-cycled cryostat and measured diffraction peaks using a four-circle geometry and $\lambda$ = 2.305 Å neutrons produced via the (002) plane of pyrolytic graphite monochromator. At HB-1A, two CeTi$_3$Bi$_4$ crystals, oriented in the [H, 0, L] and [0, K, L] scattering plane respectively, were measured in a liquid helium cryostat using neutron wavelength of $\lambda$ = 2.38 Å selected by a double-bounce pyrolitic-graphite (PG) monochromator system and a PG analyzer using PG (0 0 2).

We conducted single-crystal neutron diffraction under a magnetic field using the WAND$^2$ diffractometer at HFIR and the thermal triple-

axis spectrometer TAS-2 at the research reactor JRR-3 of the Japan Atomic Energy Agency (JAEA). At WAND[2], the sample was aligned in the Mag-B vertical cryomagnet ($0 < H < 4.5$ T) and measured with the (0KL) plane aligned horizontally. We used an incident neutron wavelength of $\lambda = 1.48$ Å. For the TAS-2 experiment, the sample was loaded into a cryogen-free 10 T magnet to have the (0KL) horizontal scattering plane. The spectrometer was fixed to the elastic condition in a triple-axis mode with a wavelength of 2.36 Å. The collimation set of open-40′-40′-40′ was employed together with a PG filter placed before the sample.

The rocking curves measured in all three diffraction experiments exhibit nearly resolution-limited Bragg peak linewidths (0.25°–0.35°), as shown in Supplementary Fig. 2.

### Angle-resolved photoemission spectroscopy (ARPES)

Synchrotron-based ARPES measurements were performed at the Lawrence Berkeley National Laboratory's Advanced Light Source (ALS) beamline 10.0.1.1 using a Scienta R4000 hemispherical electron analyzer. The ARPES spectra presented in the main text were measured using 65 eV of incident photon energy. The angular and energy resolutions were set to 0.2° and 15 meV, respectively. Measurements were conducted at 20 K under ultra-high vacuum conditions ($-10^{-11}$ Torr). High-quality single crystals were cut and mounted on a copper post using silver epoxy paste, and ceramic posts were affixed to the samples. To prevent oxidation, all sample preparation took place in a glove box. After preparation, the samples were loaded into the main chamber, cooled, and pumped down for several hours before measurements.

As described in the main text, we used pseudo-labels of high-symmetry **Q** points from a hexagonal framework for the ARPES data analysis. Despite the orthorhombic symmetry, the in-plane distortion of the Ti kagome lattice is marginal, as indicated by the lattice parameter ratio $\frac{b}{a} = 1.741 \simeq \sqrt{3}$ (see Supplementary Table 1). Thus, using high-symmetry **Q** points from a hexagonal framework remains effective in mapping the kagome band structure features onto the orthorhombic reciprocal space of CeTi$_3$Bi$_4$. The pseudo-hexagonal nature of the material is especially relevant to ARPES since the states at $E_F$ are derived predominantly from the nearly hexagonal kagome network (the orthorhombic character is introduced by the Ce$^{3+}$ chain configuration, see Fig. 1a). As such, we use **Q** labels projected onto the quasi-hexagonal 2D reciprocal space (denoted with an overline, e.g., $\bar{M}$ in Fig. 4a) for the ARPES results.

### Density functional theory (DFT) calculations

To understand the purely electronic features of the CeTi$_3$Bi$_4$ system, we examine the DFT calculated band structure for LaTi$_3$Bi$_4$. Utilizing the La-congener instead of CeTi$_3$Bi$_4$ significantly decreases the complexity originating from the partially filled Ce 4f shell, but yields qualitatively identical results around the Fermi energy as the Ln 4f orbitals are far from it. Indeed, consistent with many smaller kagome compounds (e.g., CsV$_3$Sb$_5$[46] and ScV$_6$Sn$_6$[47]), the electronic structure of LnTi$_3$Bi$_4$ near $E_F$ is almost entirely dominated by the Ti kagome orbitals.

The band structure of LaTi$_3$Bi$_4$ has been calculated within DFT based approach[48]. The electronic exchange-correlations were described using generalized gradient approximation and the Perdew–Burke–Ernzerhof[49] parametrization. The calculations were executed using the plane-wave basis projector augmented-wave approach[50] as implemented in the Vienna Ab-initio Simulation Package (VASP)[51,52]. A plane-wave energy cutoff of 450 eV has been used. A $12 \times 12 \times 12$ k-points mesh was used to execute self-consistency calculations, while the Fermi surface was calculated with a $20 \times 20 \times 20$ k-points mesh. The original structure atomic positions were optimized until the energy change does not exceed 0.05 meV and forces 0.5 meV/A. The .cif file of the optimized crystal structure is provided in Supplementary Data 1. To build the Fermi surface, the modular code

xsfconverter [https://github.com/jenskunstmann/xsfconvert] together with FermiSurfer[53] and XCrySDen[54] graphical packages.

## Data availability

The source data used in Figs. 2e, 3c, d, and 3f–h are available in the figshare database under the accession code https://doi.org/10.6084/m9.figshare.28728692. The data supporting the findings of this study are available within the paper and the Supplementary Information. Further raw data are available from the corresponding author upon request.

## Code availability

Custom codes used in this article are available from the corresponding authors upon request.

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

## Acknowledgements

We acknowledge O. Garlea for helpful discussions and thank Sung Kwan Mo for beamline assistance at the Advanced Light Source (ALS), Lawrence Berkeley National Laboratory. This work was supported by the U.S. Department of Energy, Office of Science, Basic Energy Sciences, Materials Sciences and Engineering Division. This research used resources at the High Flux Isotope Reactor, a DOE Office of Science User Facility operated by the Oak Ridge National Laboratory. The beam time was allocated to HB-1A on proposal number IPTS-32021.1 and WAND[2] on Proposal No. IPTS-32078.1. This work is based on experiments performed at the Swiss spallation neutron source SINQ, Paul Scherrer Institute, Villigen, Switzerland. M.N. acknowledges the support from the US Department of Energy (DOE), Office of Science, Basic Energy Sciences grant number DE-SC0024304 and the Air Force Office of Scientific Research MURI (Grant No. FA9550-20-1-0322). The TAS-2 experiment at JRR-3 was performed under the US–Japan Cooperative Program on Neutron Scattering. This research used resources of the Advanced Light Source, a U.S. Department of Energy Office of Science User Facility, under Contract No. DE-AC02-05CH11231. This research used resources of the National Energy Research Scientific Computing Center (NERSC), a Department of Energy Office of Science User Facility. Notice: This manuscript has been authored by UT-Battelle, LLC, under contract DE-AC05-00OR22725 with the US Department of Energy (DOE). The US government retains and the publisher, by accepting the article for publication, acknowledges that the US government retains a non-exclusive, paid-up, irrevocable, worldwide license to publish or reproduce the published form of this manuscript or allow others to do so, for US government purposes. DOE will provide public access to these results of federally sponsored research in accordance with the DOE Public Access Plan (https://www.energy.gov/doe-public-access-plan).

## Author contributions

P.P., B.R.O., and A.D.C. conceived the project. B.R.O. synthesized the samples and measured the bulk properties. P.P., S.A.C., M.D.F., W.T., R.S., D.G.M., C.T., K.K., W.T., L.M.D.-S., M.B.S., and A.D.C. conducted the neutron diffraction measurements. P.P. analyzed the neutron diffraction data. M.S., A.P.S., and M.N. conducted ARPES measurements. D.S.P. and G.D.S. performed the DFT calculations. H.M. contributed to the

theoretical interpretation and discussion. P.P., B.R.O., and A.D.C. wrote the paper with contributions from all authors.

## Competing interests

The authors declare no competing interests.
