## [Transparent Peer Review file · Nature Communications]

Spin density wave and van Hove singularity in the kagome metal CeTi₃Bi₄

Corresponding Author: Dr Pyeongjae Park

Version 0:

Reviewer comments:

Reviewer #1

(Remarks to the Author)

The work entitled "Spin density wave and van Hove singularity in the kagome metal CeTi₃Bi₄" aims at establishing a relation between the stabilization of an incommensurate spin density wave (SDW) phase in CeTi₃Bi₄ and electronic instabilities generated by van Hove singularities in the electronic band structure of the material.

The paper opens with a discussion about electronic driven charge density waves (CDW) in metallic Kagome materials. It then brings into question whether or not a SDW phase can also be stabilized by this same kind of electronic instabilities via a coupling of the electronic spin and charge degrees of freedom. Naturally, the purpose of the paper is to show that such phenomenology actually takes place in CeTi₃Bi₄. Multiple experimental techniques are employed and discussed.

The choice of CeTi₃Bi₄ is well motivated. As a Kagome-type material (Ti sublattice) hosting a sub lattice of localized Ce 4f states, CeTi₃Bi₄ is indeed a good candidate to that.

The first set of measurements establishes the macroscopic characterization of the material. Heat capacity and magnetization measurements (including the magnetic anisotropy) are discussed. It is concluded that Ce 4f electrons are well localized (negligible Kondo hybridization). This will be in turn relevant during the discussion of the electronic band structure.

Then, extensive neutron diffraction experiments and analysis are presented and establish 3 important results: a) the onset of a commensurate AFM order (order of the Ce derived local moments), b) the onset of a long wave length incommensurate AFM order (that is associated with an incommensurate SDW) and c) the resulting magnetic structure in the presence of the just described coexisting AFM orders. Some specific properties of the interaction between the two orders are discussed by detailed data and analysis.

Motivated by these results, the electronic band structure of the material is investigated by ARPES and by DFT calculations (the LaTi₃Bi₄ is adopted as a proxy for the uncorrelated electronic structure). It is asserted that ARPES establishes the presence of the van Hove singularities (vHs) close to the M points and, most striking, that the Q vector of the SDW order connects distinct (vHs) in the Fermi surface, suggesting a strong link between the vHs and the SDW order.

Overall, I found the paper clearly written and well organized. Results are sound and certainly may warrant publication in Nature Comm. Before I can make a conclusive recommendation, however, I would like some clarification about the following points in the manuscript.

Introduction

1)

The authors write: "(...) nesting instability responsible for CDW formation could be quenched by spin degrees of freedom(...)".

I am having some difficulties with the wording in this sentences. I went to the cited Refs and could not really understand what is meant by "quenched" here. Could the authors elaborate on what they mean and maybe change the wording in the manuscript for more precision and clarity?

Single Crystal Neutron Diffraction

2)

The authors write:

"(...)these incommensurate reflections were consistently observed across multiple measurements using three different crystals and diffractometers(...)"

When writing this, the authors are providing an argument to support the claim that two observed AFM are intrinsic to the material and do not stems, for instance, from mosaics and other structural imperfections.

While I accept the argument as it is, I think that analyzing the broadening of rocking curves (X-ray data) can make a more compelling case and would make the paper more complete. So, if this data set can be obtained in a timely manner, I would suggest that at least one or two reflections could be investigated.

Electronic structure analysis

3)

Figure 4

I suggest that the band labels adopted in the supplemental material should be included in figure 4 and discussed in the main text.

I also think that figure 4c makes a bad case for "good description" of the ARPES data by the DFT calculations. Indeed, I found this case much better presented by the Fermi surface figure in the supplemental material. It was only inspecting FigS9 that I was convinced that ARPES was showing the beta band. Maybe one should play around with the color scale and try to make the beta band more bright in figure 4c. Maybe the FS in figS9 could be in the main text showing the beta band.

Moreover, I would like to know (and I suggest that it should be included in the relevant figure captions) which was the energy "integration" interval that was considered to construct the Fermi surfaces in the paper.

4)

The authors associate Van hove singularities to the bright spot in figure 4c close to the M point, but ARPES intensities are under the influence of many variables. Indeed, the use of symmetrized data is sometimes a necessity because of geometric parameters, etc, etc. Is it possible to give a more quantitative analysis showing that the probed electronic band structure is approaching an inflection point close to M? Maybe a quantitative analysis of the curvature of the band structure in the vicinity of M can provide that.

5)

The authors write "(...) we utilized the result from LaTi3Bi4 as it allows for investigating the purely electronic features of (...)"

I take that electronic correlations are "purely electronic effects"... while utilizing the DFT calculating electronic structure for the La-based materials one is, to a certain extent, investigated the uncorrelated electronic structure . I think that this is more precise.

6)

Heat capacity analysis

The fact that gamma is small is important since it crosschecks with the assumption made about the relation between the electronic structures of the La- and Ce-based materials.

In this regard, I suggest that only a comparison between gamma obtained for CeTi3Bi4 and for LaTi3Bi4 can indeed exclude any relevant mass enhancement in the Ce-based material. Comparing experimental gamma for CeTi3Bi4 with DFT-obtained gamma for LaTi3Bi4 can also be accepted, although experimental values are better.

Moreover, I think that 35 mJ/mol (or about 4.3 if a per atom normalization is included) while not a large number do not

exclude some mass enhancement, and therefore I would like to see at least the comparison with the number obtained from DFT.

I think it is important to establish that indeed correlations effects are weak in CeTi₃Bi₄, since in this case features of the uncorrelated electronic structure (VHSs, nesting, etc) are likely to dominate the Physics of the material, which is the scenario proposed by the authors in the CeTi₃Bi₄ case.

7)

Magnetization measurements

Can the authors confirm that the M(H) were indeed measured along a? Why was that? (sample morphology, maybe).

I ask the same about the M/H measurements in figure 1.

I also suggest that the authors could compare Cp measurements with $d(T \times \chi_a)/dT$ to exactly pinpoint the position of the AFM ordering temperature (see M. E. Fisher, The Philosophical Magazine: A Journal of Theoretical Experimental and Applied Physics 7, 1731 (1962)).

Moreover, I find it confusing to label the high field spin polarized phase a "Polarized FM phase" (figure 3h and supplemental). For instance, do the authors suggest that indeed Cp in high field would reveal a phase transition when cooling? Why not simple a spin polarized high field phase?

Reviewer #2

(Remarks to the Author)

The manuscript "Spin density wave and van Hove singularity in the kagome metal CeBi₃Ti₄" provides a detailed study of the magnetic properties of the kagome-layered material CeBi₃Ti₄. This is a timely study, both due to its impact on understanding the physics of kagome metals, which have recently received a lot of attention, but also for its implications for understanding competing orders near van Hove singularities.

Among the main findings is a temperature-magnetic field phase diagram exhibiting both an incommensurate magnetic phase and a phase where incommensurate and commensurate magnetism coexists. These findings originate from a combination of magnetization measurements, neutron diffraction measurements, and ARPES.

The neutron diffraction measurements were conducted on separate diffractometers using different crystals which, in my opinion, provides strong evidence that the incommensurate magnetism observed originates from CeBi₃Ti₄ and is not a measurement effect [Fig. 2(a)-(e)]. The ARPES results are backed up by DFT calculations (for technical reasons these are for LaBi₃Ti₄) and there is overall agreement between the experimental and theoretical Fermi surfaces [Fig. 4(g)]. The incommensurate magnetic order is found to be unidirectional and is thus attributed to a modulation of the magnitude of the local magnetic moment on the Ce atoms arising because of RKKY interactions. In contrast, the commensurate magnetic order is a single-Q antiferromagnetic order. Both are consistent with ARPES/DFT results which show van Hove singularities at the M points nested by q_{ic} and q_c (the incommensurate and commensurate wave-vectors).

Overall, this is a well-written manuscript, and the results are clearly supported by the experimental and theoretical data presented. The findings are interesting and should engender further studies on the interplay between localized moments and nesting instabilities promoted by van Hove singularities. I therefore recommend the manuscript be published once the authors have addressed the few minor comments I have:

1) Perhaps I missed it, but how was the incommensurate order assigned to the Ce atoms?

2) While the ARPES and DFT results agree reasonably well at the level of the Fermi surface, as shown in Fig. 4(f) and (g), the comparison between the DFT-obtained bands and the measured bands [Fig. 4(c)] is less convincing [This is also evident by the large Gamma-centered pocket in Fig. 4(g) which is much smaller in Fig. 4(f)]. The results agree near the vHS so this is unlikely to affect the conclusions, but could the authors comment as to likely reasons for this disagreement?

Reviewer #3

(Remarks to the Author)

P. Park et al., present an original work addressing the connection between the spin density wave (SDW) instability and the van Hove singularity (VHS) nesting on the Fermi surface in a rare-earth based magnetic kagome compound CeTi₃Bi₄; a material class that has attracted significant attention recently.

In particular, the authors take a novel approach consisting in investigating a sample where the metallic and the magnetic properties arise from different frameworks within the crystal lattice. Specifically, the zig-zag Ce³⁺ chains, with a $J_{eff}=1/2$ ground state, are responsible for the magnetic response, while the Ti-metallic kagome planes primarily contribute to the Fermi surface. This represents an original strategy in view of the existing literature where magnetic ions are typically part of the kagome lattice.

The authors provide results from both single-crystal neutron diffraction and angle resolved photoemission spectroscopy

(ARPES), supported by DFT calculations, to explore the intertwined magnetic and electronic properties of the system. Along with these techniques, they conduct additional characterizations such as magnetization and specific heat measurements. Neutron diffraction reveals the occurrence of a commensurate antiferromagnetic (AFM) order below $T_N=3.0\text{K}$, with scattering at $Q_C=Q=Q_{Nuc}+(0,1,0)$ and an incommensurate phase, with a characteristic magnetic scattering wavevector $Q_{IC}=Q_{Nuc}+(0,0.94,0)$ below $T_2=3.4\text{K}$, persisting below T_N . After considering several models, the authors interpret the incommensurate phase as a spin density wave (SDW) modulation with a long period of 17.5 nm, characterized by modulated and antiferromagnetically interacting Ce^{3+} magnetic moments along the b-axis (inter-chain direction) confirming a strong easy-axis anisotropy, as seen in the magnetization data.

The SDW is thus found to be more robust against thermal fluctuations and magnetic field as compared to the AFM commensurate order, with a larger stability range in terms of temperature and magnetic field amplitude. At higher fields (temperatures), the SDW fades into a ferromagnetic field polarized phase (paramagnetic phase). The authors state that while the co-existence or phase segregation of these states remains an open question, the locking of the incommensurability parameter to 0.94 below T_N rather supports a picture of interdependent magnetic states.

The ARPES data, supported by DFT calculations, reveal the existence of van Hove singularities at the M points. The Energy vs k maps demonstrate the saddle like nature of M with the occurrence of a high density of states (DOS) around M. The Fermi surface and DOS further demonstrate that Q_C and Q_{IC} connect the DOS at and around the M points (with a nesting vector aligned parallel to b^*), respectively. While RKKY-mediated magnetic interactions dominate the occurrence of the two magnetic states, they are unable to account for the persistence of the SDW below T_N . The authors thus suggest that the conduction electrons notably contribute to the SDW through the VHS nesting instability that may drive the SDW. This manuscript tackles a timely topic with convincing data and clear methodology. The paper is well written, with step-by-step conclusions that contribute new insights into the physics of kagome metals. I strongly recommend it for publication in Nature Communications after the authors address few questions:

1. The authors mention the relevance of VHS to the occurrence of CDW in kagome metals, as demonstrated in earlier reports (Ref 7,8,10, 12-20).

In the last paragraph of page 3, I suggest the statement “Their modulation wave vectors are nearly identical to that associated with the $2a \times 2a$ CDW found in other kagome metals ” is completed by a statement like “and where the VHS near the Fermi level has been shown to be relevant for the occurrence of CDW”. Indeed, since no CDW was reported so far in CeTi_3Bi_4 , the main connection between those systems is the occurrence of VHS.

2. On cooling, the incommensurability parameter δ varies with temperature before locking at $\delta=0.94$ below T_N . This indeed suggests that both the commensurate AFM order and the SDW and linked favouring the picture of a double-Q magnetic instability below T_N . Along this line of thought, the following questions arise and would be interesting to discuss to eventually bring further support to this scenario :

- Did the authors observe any temperature dependent variation of the SDW correlation lengths from the single-crystal neutron diffraction data?
- Is it possible to tell from the current data whether the SDW and commensurate phases exhibit the same correlation lengths or whether they are both Q-resolution limited?
- The magnetic moment (Supp. Table 4) extracted from the ZEBRA and WAND² data is only given at 1.5K. Could the authors provide an estimate of the SDW averaged magnetic moment amplitude above T_N ?
- In the same spirit, could the author please provide the raw (non-scaled) data of Fig.8 of the supplementary material?

3. From the WAND² data Fig.2.d and from Fig.3, it seems like there is some diffuse scattering around the magnetic peaks in the $[0K0]/[00L]$, and also near the nuclear Bragg spots (Fig.1.a-b) in the $[H00]/[00L]$ scattering plane. Do the authors please have an interpretation? Is this diffuse scattering temperature dependent or could it result from sample texture, lattice planes stacking or instrumental resolution, for instance?

4. Regarding the ARPES data, the authors attribute the electron pocket labelled α to surface states near the Γ point, as it is absent on the DFT calculations.

The surface state nature of the electronic bands can usually be checked experimentally by varying the incident energy and looking at whether the binding energy of the bands varies or not. Did the author perform such a test during the beamtime? Otherwise, although the Fermi surface is mainly due to Ti-metallic layers, and since the DFT calculations are based on the non-magnetic La instead of Ce, which of course is well justified within the manuscript, could there be any other potential origin for the α band?

5. Could the authors please specify the incident energy used for the ARPES measurements in the Methods section?

6. Also, could the author please clarify what they mean by “ a standard reference frame of the primitive orthorhombic face-centered cubic (FCC) lattice” in Supplementary Note 1 ?

I have also minor comments:

7. In the abstract: “These findings establish the rare-earth kagome metals LnTi_3Bi_4 as a model platform where the characteristic electronic structure of the kagome lattice plays a pivotal role in magnetic order”.

8. Last paragraph of page 3, there is a repetition: “Interestingly, ARPES and DFT results identify VHSs near E_f at the M points of the pseudo-hexagonal reciprocal lattice, manifesting high DOS around the M points”.

9. Page 10 of the Supp Mat (Table 5 instead of 4. One the same table, “Whychoff letter” should be replaced by “Wyckoff positions”).

10. Fig.3, Supp Fig.6-7 : instrument name missing on the figure and the caption.

11. Fig.2.e, the error bars seem missing. If smaller than the point size, then please indicate it.

Version 1:

Reviewer comments:

Reviewer #1

(Remarks to the Author)

All my points of concern were answered by the authors in a very professional and comprehensive manner. I strongly recommend the publication of the manuscript in its present form.

Reviewer #2

(Remarks to the Author)

The authors have provided satisfactory answers to my questions. I recommend publication of the manuscript.

Reviewer #3

(Remarks to the Author)

The authors have thoroughly responded to all my inquiries and made the appropriate revisions in both the main text and the supplementary material. I thus have no further comments and fully support the publication of the manuscript in its current form in Nature Communications.

We thank all three referees for their thoughtful reviews and supportive comments regarding our work, as well as their positive assessment of its suitability for publication in *Nature Communications*. We appreciate their recognition of the timeliness and significance of our study in addressing the interplay between the van Hove singularity in kagome metals and magnetism. Notably, all referees have acknowledged that our work convincingly demonstrates the presence of this coupling in CeTi₃Bi₄. Below, we provide a point-by-point response to their comments, from which we could further enhance the clarity of our manuscript.

First Report of Referee 1

General Comment: The work entitled “Spin density wave and van Hove singularity in the kagome metal CeTi₃Bi₄” aims at establishing a relation between the stabilization of an incommensurate spin density wave (SDW) phase in CeTi₃Bi₄ and electronic instabilities generated by van Hove singularities in the electronic band structure of the material.

The paper opens with a discussion about electronic driven charge density waves (CDW) in metallic Kagome materials. It then brings into question whether or not a SDW phase can also be stabilized by this same kind of electronic instabilities via a coupling of the electronic spin and charge degrees of freedom. Naturally, the purpose of the paper is to show that such phenomenology actually takes place in CeTi₃Bi₄. Multiple experimental techniques are employed and discussed.

The choice of CeTi₃Bi₄ is well motivated. As a Kagome-type material (Ti sublattice) hosting a sub lattice of localized Ce 4f states, CeTi₃Bi₄ is indeed a good candidate to that.

The first set of measurements establishes the macroscopic characterization of the material. Heat capacity and magnetization measurements (including the magnetic anisotropy) are discussed. It is concluded that Ce 4f electrons are well localized (negligible Kondo hybridization). This will be in turn relevant during the discussion of the electronic band structure.

Then, extensive neutron diffraction experiments and analysis are presented and establish 3 important results: a) the onset of a commensurate AFM order (order of the Ce derived local moments), b) the onset of a long wave length incommensurate AFM order (that is associated with an incommensurate SDW) and c) the resulting magnetic structure in the presence of the just described coexisting AFM orders. Some specific properties of the interaction between the two orders are discussed by detailed data and analysis.

Motivated by these results, the electronic band structure of the material is investigated by ARPES and by DFT calculations (the LaTi₃Bi₄ is adopted as a proxy for the uncorrelated electronic structure). It is asserted that ARPES establishes the presence of the van Hove singularities (vHs) close to the M points and, most striking, that the Q vector of the SDW order connects distinct (vHs) in the Fermi surface, suggesting a

strong link between the vHs and the SDW order.

Overall, I found the paper clearly written and well organized. Results are sound and certainly may warrant publication in Nature Comm. Before I can make a conclusive recommendation, however, I would like some clarification about the following points in the manuscript.

Reply: We appreciate the referee's positive view on our work and their constructive comments. We hope that the point-by-point responses below have satisfactorily addressed the referee's questions and concerns.

Comment #1: Introduction, 1) The authors write: "(.) nesting instability responsible for CDW formation could be quenched by spin degrees of freedom(...)".

I am having some difficulties with the wording in this sentences. I went to the cited Refs and could not really understand what is meant by "quenched" here. Could the authors elaborate on what they mean and maybe change the wording in the manuscript for more precision and clarity?

Reply: We apologize for any confusion caused by our choice of wording. To enhance clarity, we have simplified the sentence in the manuscript to read:

"...a nesting instability between VHSs could lead to diverging spin-spin correlations, resulting in a spin-density wave (SDW) state—a modulation of the magnetic moments—analogue to how the instability induces CDW through diverging charge correlations."

Regarding the referee's question about the term "**quenched**" in the original wording, our intended meaning was that a Fermi surface **instability** associated with a specific nesting wave vector (or periodicity in real-space terms) can be "**resolved**" by the emergence of a modulation in either the charge degrees of freedom (charge density wave, CDW) or the spin degrees of freedom (spin density wave, SDW).

Comment #2: Single Crystal Neutron Diffraction, 2) The authors write: "(...)these incommensurate reflections were consistently observed across multiple measurements using three different crystals and diffractometers(...)"

When writing this, the authors are providing an argument to support the claim that two observed AFM are intrinsic to the material and do not stems, for instance, from mosaics and other structural imperfections.

While I accept the argument as it is, I think that analyzing the broadening of rocking curves (X-ray data) can make a more compelling case and would make the paper more complete. So, if this data set can be obtained in a timely manner, I would suggest that at least one or two reflections could be investigated.

Reply: Recognizing the referee’s concern, we have conducted additional analysis on the rocking curves of nuclear and magnetic Bragg peaks obtained from our three “neutron” diffraction measurements. This analysis is now included in the Supplementary Information as a new figure (Supplementary Fig. 2 in the revised version, or see Fig. R1 below). The measured rocking curves exhibit full width at half maximum (FWHM) values of 0.25° – 0.35° consistently across all three independent measurements. We emphasize that these values are nearly resolution-limited for neutron diffractometers. We also note that the commensurate (Q_C) and incommensurate (Q_{IC}) reflections appear only below $T_2 = 3$ K and $T_N = 3.4$ K (see Fig. 2e), coinciding with the magnetic phase transition identified through bulk characterizations (Fig. 1d–e). Additionally, there is no appreciable change in the width of the nuclear peaks (see Fig. R1 below). Taken together, these points reaffirm the quality of our measured crystals and further rule out extrinsic factors such as mosaicity or structural imperfections as potential contributors to the observed magnetic reflections.

We note that X-ray diffraction (XRD) rocking curve analysis is not feasible due to several practical limitations. First, $CeTi_3Bi_4$ becomes radioactive after neutron irradiation. Thus, the samples used in the neutron diffraction experiments cannot be cleared for release and re-examined using single-crystal XRD in our laboratories. Second, the size of crystals used in our neutron experiments (5–10 mm) is significantly larger than both the typical X-ray beam spot and penetration depth and thus would not provide a comprehensive assessment of the overall mosaicity and crystal quality. Third, single-crystal XRD measurements at temperatures below 3 K ($< T_N$) are challenging and not available in our laboratories. In contrast, the neutron diffraction analysis discussed in the preceding paragraph and the new supplementary figure largely overcomes these challenges.

Fig. R1. Three crystals and corresponding rocking curves of nuclear and magnetic Bragg peaks measured at three different neutron beamlines. The angle values in each panel indicate the full width at half maximum (FWHM) of the respective peaks. For the nuclear peaks measured at ZEBRA and WAND², error bars (= standard deviations) are much smaller than the data symbols. All samples exhibit near-resolution-limited FWHM, confirming high sample quality and ruling out extrinsic factors as the source of the observed commensurate and incommensurate magnetic reflections. For the magnetic reflection measurements at TAS-2, a rocking scan was not performed; instead, a $[0, K, 0]$ scan is shown in the bottom-right panel. The approximate FWHM in azimuthal rotation angle (ω) was estimated based on its relation to the b^* vector and may not be directly comparable to the FWHM values obtained from rocking scans of the nuclear peaks.

Comment #3: Electronic structure analysis, 3) Figure 4, I suggest that the band labels adopted in the supplemental material should be included in figure 4 and discussed in the main text.

I also think that figure 4c makes a bad case for “good description” of the ARPES data by the DFT calculations. Indeed, I found this case much better presented by the Fermi surface figure in the supplemental material. It was only inspecting FigS9 that I was convinced that ARPES was showing the beta band. Maybe one should play around with the color scale and try to make the beta band more bright in figure 4c. Maybe the FS in figS9 could be in the main text showing the beta band.

Moreover, I would like to know (and I suggest that it should be included in the relevant figure captions) which was the energy "integration" interval that was considered to construct the Fermi surfaces in the paper.

Reply: We thank the referee for the nice suggestions. In response, we have incorporated the band labels (α , β , γ , δ , ζ) from the Supplementary Information into Fig. 4 of the main text. We have also replaced the original Fig. 4f (ARPES intensity) with the Fermi surface plot previously shown in Supplementary Fig. 9 (the second derivative of ARPES intensity). This modification provides a clearer visualization of the Fermiology of CeTi₃Bi₄, including the β -band.

Finally, we did not integrate the spectrum beyond the kinetic energy step size, which was set to 5 meV. Thus, the integration width is 5 meV, smaller than the energy resolution of 15 meV. We have specified the energy integration range used to construct our Fermi surface plots in the caption of Fig. 4.

Comment #4: 4) The authors associate Van hove singularities to the bright spot in figure 4c close to the M point, but ARPES intensities are under the influence of many variables. Indeed, the use of symmetrized data is sometimes a necessity because of geometric parameters, etc, etc. Is it possible to give a more quantitative analysis showing that the probed electronic band structure is approaching an inflection point close to M? Maybe a quantitative analysis of the curvature of the band structure in the vicinity of M can provide that.

Reply: Following the referee’s suggestion, we have conducted a more careful analysis of the band structure around the \bar{M}' point. The results are now included in the revised manuscript as Supplementary Fig. 12 (also shown in Fig. R2 below).

All three panels clearly demonstrate that the concave parabolic dispersion along the k_x direction ($\mathbf{k} \parallel \overline{\Gamma M'}$) shifts toward higher energy as one moves away from the \bar{M}' point along the k_y direction ($\mathbf{k} \parallel \overline{M'K}$). This systematic shift provides direct visualization of the saddle point nature of the dispersion at the \bar{M}' points, i.e., a van Hove singularity. To illustrate this more explicitly, panel **a** marks the momentum positions where the dispersion crosses the Fermi energy using two sets of data symbols “+” and “o”, each representing positions extracted from fitting of the momentum distribution curves (MDCs) of the photoemission intensity and its second derivative along the two principal momentum directions.

Fig. R2. Detailed analysis of the band structure around the \bar{M}' point. The color scale in **a** represents photoemission intensity, while **b** shows the same intensity distribution but symmetrized with respect to the $\overline{KM'K}$ line.

Comment #5: The authors write “(...) we utilized the result from LaTi3Bi4 as it allows for investigating the purely electronic features of (...)”

I take that electronic correlations are “purely electronic effects”... while utilizing the DFT calculating electronic structure for the La-based materials one is, to a certain extent, investigated the uncorrelated electronic structure. I think that this is more precise.

Reply: We acknowledge that our original wording could have inadvertently suggested that our DFT calculations on LaTi_3Bi_4 capture “electronic correlation effects”. We have revised the sentence for clarity:

“...as it allows for investigating the uncorrelated electronic structure of LnTi_3Bi_4 near the Fermi level with significantly decreased complexity of the calculation due to the non-magnetic nature of La”.

In our previous text, the phrase “**purely electronic** features” was intended to convey that the calculation **does not include the magnetism** arising from the partially filled Ce^{3+} 4f shell, whose spin-orbital coupled nature is challenging to capture accurately via DFT. As described in the Methods section, this simplification is well justified by the fact the electronic structure of LnTi_3Bi_4 near the Fermi energy is predominantly determined by the Ti kagome orbitals. Consequently, replacing Ce with La still provides a reasonable estimation of the Fermi surface of CeTi_3Bi_4 , as long as electron correlation effects inherent to Ce^{3+} –Kondo effects–do not play a significant role (this is addressed in our response to Comment #6).

Indeed, the Fermiology revealed by our DFT calculations (Fig. 4g) aligns well with the ARPES results (Fig. 4f), in confirming the presence of a van Hove singularity at the M' points, which is a key finding of our electronic structure analysis.

Comment #6: 6) Heat capacity analysis. The fact that gamma is small is important since it crosschecks with the assumption made about the relation between the electronic structures of the La- and Ce-based materials.

In this regard, I suggest that only a comparison between gamma obtained for CeTi_3Bi_4 and for LaTi_3Bi_4 can indeed exclude any relevant mass enhancement in the Ce-based material. Comparing experimental gamma for CeTi_3Bi_4 with DFT-obtained gamma for LaTi_3Bi_4 can also be accepted, although experimental values are better.

Moreover, I think that 35 mJ/mol (or about 4.3 if a per atom normalization is included) while not a large number do not exclude some mass enhancement, and therefore I would like to see at least the comparison with the number obtained from DFT.

I think it is important to establish that indeed correlations effects are weak in CeTi_3Bi_4 , since in this case features of the uncorrelated electronic structure (VHSs, nesting, etc) are likely to dominate the Physics of the material, which is the scenario proposed by the authors in the CeTi_3Bi_4 case.

Reply: We thank the referee for raising this important point. To directly address it, we have measured low-temperature heat capacity of LaTi_3Bi_4 , a non-magnetic analogue of CeTi_3Bi_4 lacking potential correlation effects associated with Ce^{3+} ions. The results, shown in Fig. R3 below (now included as Supplementary Fig. 1), reveal a γ value of 10.2(1) [$\text{mJ mol}^{-1} \text{K}^{-2}$]. This value is approximately 3.4 times smaller than that of CeTi_3Bi_4 .

While this result suggests that some degree of electron correlation is present in CeTi₃Bi₄, (note that we do not claim that the correlation effects in CeTi₃Bi₄ is absent), we would like to emphasize the original context of our assertion: the correlation effects in CeTi₃Bi₄ are not large enough to be considered dominant in determining its physical properties, unlike typical heavy-fermion Ce³⁺ systems. The γ value of CeTi₃Bi₄ is still markedly smaller than the typical values expected for heavy-fermion Ce³⁺ systems, which are generally at least $\gamma \simeq 100$ [mJ mol⁻¹ K⁻²] or even higher (see Ref. 40). The obtained γ value is also not an order-of-magnitude enhancement over the uncorrelated case (10.2(1) [mJ mol⁻¹ K⁻²]).

Beyond the γ value, we would also like to point out the additional arguments presented in Supplementary Note 4. Primarily, in heavy-fermion systems with strong Kondo temperatures (i.e., significant electron correlations), the full magnetic entropy $R \ln 2$ is typically restored at temperatures much higher than T_N (usually at $T > 10T_N$). However, in CeTi₃Bi₄, our heat capacity data (Fig. 1d) indicates the nearly full restoration around T_N , suggesting a vanishingly low Kondo temperature, inconsistent with a strongly correlated heavy-fermion system. Also, the Rietveld refinement analysis reveals a robust magnetic order in CeTi₃Bi₄ (i.e., an ordered moment's magnitude is comparable to the full moment's size estimated by saturation magnetization; see Supplementary Note 3), whereas heavy-fermion systems with strong Kondo effects typically exhibit very weak magnetic order.

Taken together with the newly added heat capacity data for LaTi₃Bi₄, we believe these experimental facts provide a strong case that Kondo effects in CeTi₃Bi₄ are marginal and not significant enough to predominantly drive the observed spin-density wave modulations. We have incorporated this discussion into Supplementary Note 4.

Fig. R3. Measured heat capacity of LaTi₃Bi₄.

Comment #7: 7) Magnetization measurements. Can the authors confirm that the $M(H)$ were indeed measured along a ? Why was that? (sample morphology, maybe).

I ask the same about the M/H measurements in figure 1.

I also suggest that the authors could compare C_p measurements with $d(T \times \chi_a)/dT$ to exactly pinpoint the position of the AFM ordering temperature (see M. E. Fisher, *The Philosophical Magazine: A Journal of Theoretical Experimental and Applied Physics* 7, 1731 (1962)).

Moreover, I find it confusing to label the high field spin polarized phase a “Polarized FM phase” (figure 3h and supplemental). For instance, do the authors suggest that indeed C_p in high field would reveal a phase transition when cooling? Why not simple a spin polarized high field phase?

Reply: Thank you for the suggestions. Below, we provide a point-by-point response to this comment:

“Can the authors confirm that the $M(H)$ were indeed measured along a ?”

We confirm that all magnetization data presented in this work were measured along the crystallographic directions as stated. The orientations were carefully determined prior to the measurements using Laue XRD, and we have added this explanation to the Methods section. This procedure was also described in Ref. 30, a previous study by the authors on $LnTi_3Bi_4$.

“I also suggest that the authors could compare C_p measurements with $d(T \times \chi_a)/dT$ to exactly pinpoint the position of the AFM ordering temperature”

Fig. R4 below shows a comparison between the C_p and $\frac{d(T\chi_a)}{dT}$ curves. Since the measured quantity in an experiment is M/H , which is equivalent to the susceptibility χ under a sufficiently small H in antiferromagnets, we have plotted $d(T \times (M/H))/dT$.

The maximum of each quantity yields $T_N = 3.15$ K and $T_N = 3.24$ K, respectively, with an average value of approximately 3.2 K. We note that slight systematic errors of 0.1 K scale may arise when comparing temperature readings from different cryostats. Thus, the small deviation of 0.1 K is well within a reasonable agreement range.

We have included this comparison into Supplementary Fig. 1 and revised T_N values in the main text for consistency.

Fig. R4. Comparison between the heat capacity and $d(TM)/dT$ quantity to locate the transition temperature.

“Moreover, I find it confusing to label the high field spin polarized phase a “Polarized FM phase” (figure 3h and supplemental). For instance, do the authors suggest that indeed C_p in high field would reveal a phase transition when cooling? Why not simple a spin polarized high field phase?”

To improve clarity, we have revised the wording to “Polarized phase”.

 First Report of Referee 2

General Comment: The manuscript “Spin density wave and van Hove singularity in the kagome metal $CeBi_3Ti_4$ ” provides a detailed study of the magnetic properties of the kagome-layered material $CeBi_3Ti_4$. This is a timely study, both due to its impact on understanding the physics of kagome metals, which have recently received a lot of attention, but also for its implications for understanding competing orders near van Hove singularities.

Among the main findings is a temperature-magnetic field phase diagram exhibiting both an incommensurate magnetic phase and a phase where incommensurate and commensurate magnetism coexists. These findings originate from a combination of magnetization measurements, neutron diffraction measurements, and ARPES.

The neutron diffraction measurements were conducted on separate diffractometers using different crystals which, in my opinion, provides strong evidence that the incommensurate magnetism observed originates from $CeBi_3Ti_4$ and is not a measurement effect [Fig. 2(a)-(e)]. The ARPES results are backed up by DFT calculations (for technical reasons these are for $LaBi_3Ti_4$) and there is overall agreement between the experimental and theoretical Fermi surfaces [Fig. 4(g)]. The incommensurate magnetic order is found to be unidirectional and is thus attributed to

a modulation of the magnitude of the local magnetic moment on the Ce atoms arising because of RKKY interactions. In contrast, the commensurate magnetic order is a single-Q antiferromagnetic order. Both are consistent with ARPES/DFT results which show van Hove singularities at the M points nested by q_{ic} and q_c (the incommensurate and commensurate wave-vectors).

Overall, this is a well-written manuscript, and the results are clearly supported by the experimental and theoretical data presented. The findings are interesting and should engender further studies on the interplay between localized moments and nesting instabilities promoted by van Hove singularities. I therefore recommend the manuscript be published once the authors have addressed the few minor comments I have:

Reply: We appreciate the referee's positive evaluation and recommendation of our work.

Comment #1: Perhaps I missed it, but how was the incommensurate order assigned to the Ce atoms?

Reply: The assignment of the incommensurate order to the Ce atoms is based on the following two key observations:

- 1) Ce ions are the only magnetic species in this material.
- 2) The incommensurate reflections observed in our diffraction experiments are of magnetic origin.

The first point is supported by heat capacity measurements, which reveal the total magnetic entropy reaching $R\ln(2)$ (i.e., $S = 1/2$) around the magnetic phase transition at ≈ 3.2 K (Fig. 1d and Ref. 30). This is exactly what is expected for Ce^{3+} magnetism, as its ground-state doublet gives an entropy corresponding to the two thermally accessible states. If any other elements in the material contributed significantly to the magnetism, it is unlikely to result in a similar saturation of magnetic entropy. Additionally, no noticeable magnetic response is observed in $LaTi_3Bi_4$, supporting the idea that Ce is the only magnetically active element in $CeTi_3Bi_4$.

The second point is supported by two experimental observations. First, the overall momentum dependence of the incommensurate Bragg peak intensities follows the behavior expected from the magnetic form factor. In other words, the incommensurate Bragg peaks weaken at high Q whereas peaks due to a structural distortion would not exhibit this dependence in neutron diffraction. Second, the incommensurate reflections emerge at 3.3~3.4 K, coinciding with the magnetic phase transition ($T_N \approx 3.2$ K) identified through bulk characterizations (Fig. 1d–e).

These arguments provide ample evidence that the incommensurate modulation observed in our neutron diffraction measurements originates from the magnetic moments on the Ce^{3+} sublattice.

Comment #2: While the ARPES and DFT results agree reasonably well at the level of the Fermi surface, as shown in Fig. 4(f) and (g), the comparison between the DFT-obtained bands and the measured bands [Fig. 4(c)] is less convincing [This is also

evident by the large Gamma-centered pocket in Fig. 4(g) which is much smaller in Fig. 4(f)]. The results agree near the vHS so this is unlikely to affect the conclusions, but could the authors comment as to likely reasons for this disagreement?

Reply: Below, we provide a two-fold response addressing 1) The discrepancy between Fig. 4f and 4g and 2) the discrepancy apparent in Fig. 4c.

1) We believe the referee's perception of an inconsistency between Fig. 4g and 4f—regarding the size of the Γ -centered pocket—was due to a mismatch between different bands (α band in Fig. 4g and β band in Fig. 4f; see the revised manuscript). This was mainly due to the weak intensity of the β band in the previous color plots of Fig. 4f, an issue also noted by Referee 1 (please see Comment # 3). We apologize for the confusion this have caused.

To improve clarity, we have revised Fig. 4f in the new version of the manuscript by i) using the second derivative plot to better show the β band, and ii) adding band labels (α , β , γ , δ , ζ) to clearly map the correspondence between Fig. 4f (ARPES) and Fig. 4g (DFT). The revised Fig. 4f now clearly reveals the β band, consistent with the “*large Gamma-centered pocket in Fig. 4g*” mentioned by the referee. Regarding the absence of the smaller Γ -centered electron pocket (α band in Fig. 4f) in the DFT calculation, we attribute this to substantial surface effects, as described in our response to the Comment # 4 of Referee 3 or Supplementary Note 7.

2) As the referee correctly pointed out, ARPES and DFT exhibit some apparent discrepancies below the Fermi energy. The primary reason should be our choice to use LaTi_3Bi_4 for the DFT calculations instead of CeTi_3Bi_4 . The reason behind this decision is—as described in the main text—the non-magnetic nature of La allows for the calculation with significantly decreased complexity. CeTi_3Bi_4 , on the other hand, features strong spin-orbital coupled magnetism of the partially filled Ce^{3+} 4f shell, which is challenging to capture accurately using DFT. This substitution does not significantly impact the electronic structure near the Fermi surface as the Ti 3d bands dominate in this region (which is why ARPES and DFT agree reasonably well at the Fermi level). Yet it introduces limitations at lower energies, where the influence of Ce^{3+} electronic states becomes more pronounced.

We have incorporated this clarification into the revised manuscript (page 7).

First Report of Referee 3

General Comment: P. Park et al., present an original work addressing the connection between the spin density wave (SDW) instability and the van Hove singularity (VHS) nesting on the Fermi surface in a rare-earth based magnetic kagome compound CeTi_3Bi_4 ; a material class that has attracted significant attention recently.

In particular, the authors take a novel approach consisting in investigating a sample where the metallic and the magnetic properties arise from different frameworks within the crystal lattice. Specifically, the zig-zag Ce^{3+} chains, with a $J_{\text{eff}}=1/2$ ground state, are responsible for the magnetic response, while the Ti-metallic kagome planes

primarily contribute to the Fermi surface. This represents an original strategy in view of the existing literature where magnetic ions are typically part of the kagome lattice. The authors provide results from both single-crystal neutron diffraction and angle resolved photoemission spectroscopy (ARPES), supported by DFT calculations, to explore the intertwined magnetic and electronic properties of the system. Along with these techniques, they conduct additional characterizations such as magnetization and specific heat measurements.

Neutron diffraction reveals the occurrence of a commensurate antiferromagnetic (AFM) order below $T_N=3.0\text{K}$, with scattering at $Q_C=Q=Q_{\text{Nuc}}+(0,1,0)$ and an incommensurate phase, with a characteristic magnetic scattering wavevector $Q_{\text{IC}}=Q_{\text{Nuc}}+(0,0.94,0)$ below $T_2=3.4\text{K}$, persisting below T_N . After considering several models, the authors interpret the incommensurate phase as a spin density wave (SDW) modulation with a long period of 17.5 nm, characterized by modulated and antiferromagnetically interacting Ce^{3+} magnetic moments along the b-axis (inter-chain direction) confirming a strong easy-axis anisotropy, as seen in the magnetization data.

The SDW is thus found to be more robust against thermal fluctuations and magnetic field as compared to the AFM commensurate order, with a larger stability range in terms of temperature and magnetic field amplitude. At higher fields (temperatures), the SDW fades into a ferromagnetic field polarized phase (paramagnetic phase). The authors state that while the co-existence or phase segregation of these states remains an open question, the locking of the incommensurability parameter to 0.94 below T_N rather supports a picture of interdependent magnetic states.

The ARPES data, supported by DFT calculations, reveal the existence of van Hove singularities at the M points. The Energy vs k maps demonstrate the saddle like nature of M with the occurrence of a high density of states (DOS) around M.

The Fermi surface and DOS further demonstrate that Q_C and Q_{IC} connect the DOS at and around the M points (with a nesting vector aligned parallel to b^*), respectively. While RKKY-mediated magnetic interactions dominate the occurrence of the two magnetic states, they are unable to account for the persistence of the SDW below T_N . The authors thus suggest that the conduction electrons notably contribute to the SDW through the VHS nesting instability that may drive the SDW.

This manuscript tackles a timely topic with convincing data and clear methodology. The paper is well written, with step-by-step conclusions that contribute new insights into the physics of kagome metals. I strongly recommend it for publication in Nature Communications after the authors address few questions:

Reply: We are grateful for the referee's appreciation of the value of our work and their strong support for the publication. We hope that the point-by-point responses below have satisfactorily addressed the referee's questions and concerns.

Comment #1: 1. The authors mention the relevance of VHS to the occurrence of CDW in kagome metals, as demonstrated in earlier reports (Ref 7,8,10, 12-20).

In the last paragraph of page 3, I suggest the statement “Their modulation wave vectors are nearly identical to that associated with the $2a \times 2a$ CDW found in other kagome metals ” is completed by a statement like “and where the VHS near the Fermi level has been shown to be relevant for the occurrence of CDW”. Indeed, since no CDW was reported so far in CeTi₃Bi₄, the main connection between those systems is the occurrence of VHS.

Reply: We appreciate the referee’s excellent suggestion to strengthen the connection between our results and previous studies on kagome metals exhibiting CDW. We have revised the sentence accordingly:

“Their modulation wave vectors are nearly identical to that associated with the $2a \times 2a$ CDW found in other kagome metals, where the VHS near the Fermi level has been suggested to be important.”

Comment #2: On cooling, the incommensurability parameter δ varies with temperature before locking at $\delta = 0.94$ below TN. This indeed suggests that both the commensurate AFM order and the SDW and linked favouring the picture of a double-Q magnetic instability below TN. Along this line of thought, the following questions arise and would be interesting to discuss to eventually bring further support to this scenario:

a. Did the authors observe any temperature dependent variation of the SDW correlation lengths from the single-crystal neutron diffraction data?

b. Is it possible to tell from the current data whether the SDW and commensurate phases exhibit the same correlation lengths or whether they are both Q-resolution limited?

c. The magnetic moment (Supp. Table 4) extracted from the ZEBRA and WAND² data is only given at 1.5K. Could the authors provide an estimate of the SDW averaged magnetic moment amplitude above TN ?

d. In the same spirit, could the author please provide the raw (non-scaled) data of Fig.8 of the supplementary material?

Reply: We appreciate the referee’s insightful suggestions. Below, we address the four comments in pairs:

a-b) Fig. R5 below shows the temperature dependence of the inverse FWHM for the $(0, \delta, -10)$ and $(0, 1, -10)$ magnetic Bragg peaks multiplied by 2π . In the absence of instrumental resolution effects, this quantity would serve as a direct measure of the correlation length for each modulation. However, we believe the instrumental resolution significantly contributes to the measured FWHM. The instrumental resolution estimated by a FWHM of nearby nuclear peaks, such as $(0, 2, -10)$, is nearly the same as that of the magnetic peaks at the base temperatures (\approx

0.011 Å⁻¹). This is further confirmed by the rocking curves shown in Supplementary Fig. 2, which exhibit nearly the same angular width for both nuclear and magnetic peaks. These observations indicate that both the commensurate and incommensurate magnetic peaks are nearly resolution-limited.

As a result, the actual correlation lengths of the magnetic peaks are expected to be significantly longer than the values inferred from Fig. R5. In such a case, attempting to correct for resolution effects through simple deconvolution would introduce substantial systematic errors. Therefore, we present the original $2\pi \times (\text{FWHM})^{-1}$ values in Fig. R5 rather than suggesting absolute correlation lengths. Instead, we focus on the qualitative temperature dependence only, summarized as follows:

- 1) No noticeable change is observed above $T_2 = 3.0$ K, but below this temperature, the correlation length of the SDW modulation gradually increases, coinciding with the onset of the commensurate spin modulation.
- 2) As the temperature further decreases below T_2 , the correlation lengths of the commensurate and incommensurate phases become markedly similar.

Given that this qualitative analysis may provide further insight into the nature of the co-existing magnetic modulations in CeTi₃Bi₄, we have incorporated these findings and relevant discussions into the Supplementary Information, specifically in Supplementary Note 6 and Fig. 8.

Fig. R5. Temperature dependence of the inverse full width at half maximum (FWHM) for the (0, δ, -10) and (0, 1, -10) magnetic Bragg peaks along the [0, K, 0] direction.

c–d) We assume that the referee intended to mean “above T_2 ” rather than “above T_N ”, as no magnetic order is present above T_N . Indeed, the dataset presented in Supplementary Fig. 8 (Supplementary Fig. 10 in the latest version) provides a reasonable estimate of the ordered moment for the pure SDW phase. Fig. R6 below displays the raw integrated intensity data from three different locations in the phase diagram without any applied scaling factor. The

corresponding maximum amplitude of the SDW moment has been added to the expanded Supplementary Table 4.

As anticipated from the intensity scale factor applied in Supplementary Fig. 10, the ordered moment of the SDW phase at 3.1 K is $\sqrt{\frac{1}{2}} = 0.71$ times the moment at 1.5 K, while at $T = 2.6$ K and $H = 2.5$ T it is $\sqrt{\frac{1}{1.67}} = 0.77$ times the moment at 1.5 K. The averaged SDW moment of each case is $2/\pi$ times the values shown in Supplementary Table 4, as described in Supplementary Notes.

Fig. R6. Integrated intensity of the $[0, \delta, L]$ incommensurate magnetic peaks at three different positions in the temperature-field phase diagram. Same as Supplementary Fig. 10, but without applying individual scale factors to match intensity scales.

Comment #3: From the WAND² data Fig.2.d and from Fig.3, it seems like there is some diffuse scattering around the magnetic peaks in the $[0K0]/[00L]$, and also near the nuclear Bragg spots (Fig.1.a-b) in the $[H00]/[00L]$ scattering plane. Do the authors please have an interpretation? Is this diffuse scattering temperature dependent or could it result from sample texture, lattice planes stacking or instrumental resolution, for instance?

Reply: We confirm that the diffuse scattering signals pointed out by the referee do not originate from the sample. Below, we provide detailed explanations.

The sharp and elongated diffuse signal around nuclear Bragg peaks, highlighted in Fig. R7, is an instrumental artifact related to the detector geometry of the neutron diffractometer, rather than an effect of sample defects or mosaicity. This is because, the elongation direction of each peak's tail signal (top panel of Fig. R7(b)) precisely aligns with the detector coverage trajectory of WAND² in reciprocal space (bottom panel of Fig. R7(b)). Such a precise coincidence is highly unlikely to be reproduced sample mosaicity or stacking faults. Moreover, this signal is temperature-independent, as it appears identical in both the right panels of both Fig. 2a and 2b.

However, we emphasize that this signal is extremely weak compared to the main Bragg peak intensity. This is evident in Fig. R7(c), which presents a one-dimensional intensity profile along the $[0, 0, L]$ direction with a finite integration range along H , covering the region where

the sharp tail signals appear. The profile shows a normal sharp peak at each Bragg index without any unusual broadening or distortion. The absence of significant diffuse scattering around Bragg peaks is also demonstrated in Supplementary Fig. 2. We note that the nuclear Bragg peak intensities in Fig. 2a–b appear strongly saturated due to the chosen color scale, as explicitly highlighted between Fig. R7(b) and R7(c). This was inevitable to visualize order-of-magnitudes weaker magnetic peaks, which makes the tiny secondary signals visually prominent in Figs. 2a–b, even though they remain negligible in magnitude.

[REDACTED]

Comment #4: Regarding the ARPES data, the authors attribute the electron pocket labelled α to surface states near the Γ point, as it is absent on the DFT calculations. The surface state nature of the electronic bands can usually be checked experimentally by varying the incident energy and looking at whether the binding energy of the bands varies or not. Did the author perform such a test during the beamtime?

Otherwise, although the Fermi surface is mainly due to Ti-metallic layers, and since the DFT calculations are based on the non-magnetic La instead of Ce, which of course is well justified within the manuscript, could there be any other potential origin for the α band?

Reply: We thank the referee for the comment, which indeed requires more explanations. In the revised manuscript, we address this issue in detail in Supplementary Note 7 and Fig. 13. Below we provide a separate explanation in response to the comment.

In addition to the datasets presented in the main text—collected at an incident photon energy of 65 eV—we further analyzed data taken at $E_i = 50, 55, 60,$ and 70 eV (Fig. R8a–b). Specifically, Fig. R8a shows the stacked momentum distribution curves (MDCs) along the surface $\bar{\Gamma}\bar{K}$ direction at the Fermi level, where the Fermi momentum of the α band (the peaks at $k \approx \pm 0.2 \text{ \AA}^{-1}$) remains unchanged. Also, Fig. R8b presents ARPES intensity plots at selected photon energies, further confirming the non-dispersive nature of the α band.

Meanwhile, it is important to note that a small electron pocket around the $\bar{\Gamma}$ point, resembling the α band, has been consistently observed in other materials from the $LnTi_3Bi_4$ family. For instance, see Refs. 30 and 38. Indeed, our DFT calculations predict an electron-like band centered at the $\bar{\Gamma}$ point, which is slightly above E_F ($E = E_F + 0.33$ eV) at $\bar{\Gamma}$ but forms an electron pocket at $k_z = \pi$. Please see red dashed circles in Fig. R8c. This is indeed a strong candidate for the observed α band.

Nevertheless, despite its possible bulk origin described above, we believe that the α band observed in ARPES possesses a strong surface character. Unlike the experimentally observed α band, our DFT calculations suggest substantial dispersion of this band along the k_z direction (the right panel of Fig. R8c). Such dispersive behavior of the $\bar{\Gamma}$ -centered pocket is found in most band structures of the $LnTi_3Bi_4$ family, which is attributed to its p -orbital character.

Thus, the α band is either a pure surface state or, a bulk-derived state that becomes localized at the surface. In both scenarios, the α band is strongly influenced by surface effects and cannot be considered an ideal bulk state.

Fig. R8. A more detailed investigation into the nature of the α band. a–b, Examination of the α band based on its dependence on incident photon energy. Panel a shows stacked momentum distribution curves (MDCs) along the surface $\bar{\Gamma}\bar{K}$ direction at the Fermi energy for photon energies ranging from 50 to 70 eV, where the α band appears as distinct peaks around $k \approx \pm 0.2 \text{ \AA}^{-1}$. Panel b presents energy-momentum maps for 50, 55, and 60 eV photon energies. c, DFT calculation results indicating a potential α band candidate, predicted to be

present above E_F . The right panel demonstrates its substantial dispersion along the out-of-plane (k_z) direction, which is inconsistent with the observed behavior in **a–b**.

Comment #5: Could the authors please specify the incident energy used for the ARPES measurements in the Methods section?

Reply: The revised main text specifies the incident energy value in Methods and Figure 4 caption, which is 65 eV.

Comment #6: Also, could the author please clarify what they mean by “ a standard reference frame of the primitive orthorhombic face-centered cubic (FCC) lattice” in Supplementary Note 1?

Reply: For “**A standard reference frame** of the primitive orthorhombic face-centered cubic (FCC) lattice”, we meant “**a standard choice of the lattice vectors** for the primitive orthorhombic face-centered cubic (FCC) lattice”. To enhance clarity, we have replaced the original terminology with the second expression in the previous sentence. We apologize for any confusion caused by our previous wording and provide a brief explanation below.

Although the **conventional** orthorhombic FCC lattice (represented by the **a**, **b**, and **c** axes in Figs. 1a–b) offers a straightforward notation, it does not represent the smallest possible unit cell of an FCC lattice. The true **primitive cell** can instead be described using three non-orthogonal lattice vectors (**a**₁, **a**₂, **a**₃), each constructed by averaging two of the three conventional lattice vectors—for example, (**a** + **b**)/2. While there are six possible ways to assign (**a** + **b**)/2, (**b** + **c**)/2, and (**c** + **a**)/2 to **a**₁, **a**₂, and **a**₃, a widely accepted **standard choice** is as follows (Eq. (3) in Supplementary Notes):

$$\mathbf{a}_1 = \left(0, \frac{b}{2}, \frac{c}{2}\right), \quad \mathbf{a}_2 = \left(\frac{a}{2}, 0, \frac{c}{2}\right), \quad \mathbf{a}_3 = \left(\frac{a}{2}, \frac{b}{2}, 0\right).$$

The relationship between these two different lattice vector notations is illustrated in Fig. R9 below.

Fig. R9. Conventional and primitive unit cell notations for the orthorhombic FCC crystal structure.

For the sake of maintaining a straightforward diffraction geometry based on orthogonal reciprocal lattice vectors, we adopted the **conventional orthorhombic** unit cell notation for our diffraction results. However, when describing the electronic structure in reciprocal space, the minimal unit (i.e., the first Brillouin zone) is properly captured only by using the primitive cell notation (\mathbf{a}_1 , \mathbf{a}_2 , \mathbf{a}_3), which leads to non-orthogonal reciprocal lattice vectors. Since this study establishes a direct connection between the ordering wave vectors observed in diffraction experiments and the nesting wave vectors in electronic structure calculations, clearly defining the relationship between these two crystallographic notations was crucial, which was the motivation of providing the Supplementary Note 1.

Comment #7: In the abstract: “These findings establish the rare-earth kagome metals LnTi_3Bi_4 as a model platform where the characteristic electronic structure of the kagome lattice plays a pivotal role in magnetic order”.

Reply: We have revised the sentence as suggested.

Comment #8: Last paragraph of page 3, there is a repetition: “Interestingly, ARPES and DFT results identify VHSs near E_f at the M points of the pseudo-hexagonal reciprocal lattice, manifesting high DOS around the M points”.

Reply: We have revised the redundant expressions in the sentence.

Comment #9: Page 10 of the Supp Mat (Table 5 instead of 4. One the same table, “Whychoff letter” should be replaced by “Wyckoff positions”).

Reply: We have corrected the typos accordingly. We again thank the referee for careful checks.

Comment #10: 10. Fig.3, Supp Fig.6-7 : instrument name missing on the figure and the caption.

Reply: We have indicated the instrument name in the figure captions. All of them were collected at WAND².

Comment #11: Fig.2.e, the error bars seem missing. If smaller than the point size, then please indicate it.

Reply: Thank you for pointing this out. We have now added missing error bars in the leftmost panel of Fig. 2e.

Second Report of Referee 1

General Comment: All my points of concern were answered by the authors in a very professional and comprehensive manner. I strongly recommend the publication of the manuscript in its present form.

Reply: We thank for the referee's strong recommendation of our work for publication.

Second Report of Referee 2

General Comment: The authors have provided satisfactory answers to my questions. I recommend publication of the manuscript.

Reply: We appreciate the referee's recommendation of our work for publication.

Second Report of Referee 3

General Comment: The authors have thoroughly responded to all my inquiries and made the appropriate revisions in both the main text and the supplementary material. I thus have no further comments and fully support the publication of the manuscript in its current form in Nature Communications.

Reply: We again thank the referee for their time in reviewing the manuscript and for their recommendation for publication.